# SAW-BENCH: Learning Situated Awareness in the Real World

Chuhan Li [1]   Rilyn Han [*2]   Joy Hsu [*3]   Yongyuan Liang [*4]
Rajiv Dhawan [5]   Jiajun Wu [3]   Ming-Hsuan Yang [6]   Xin Eric Wang [1]

https://sawbench.github.io

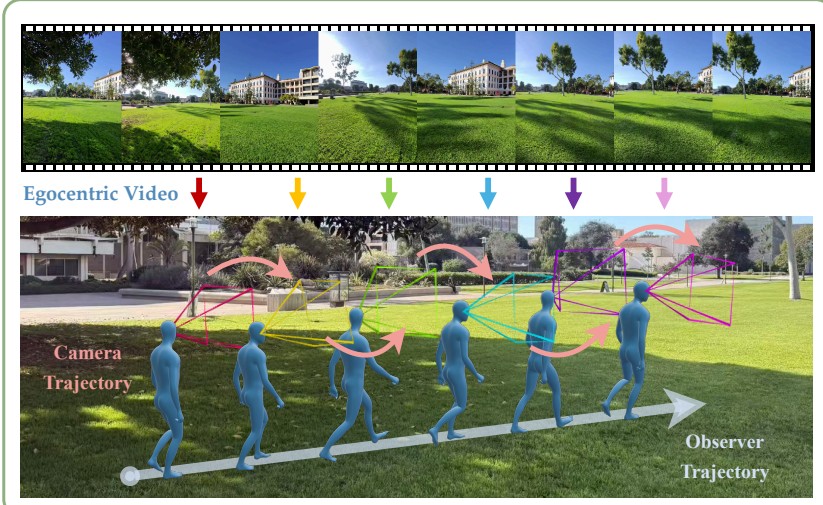
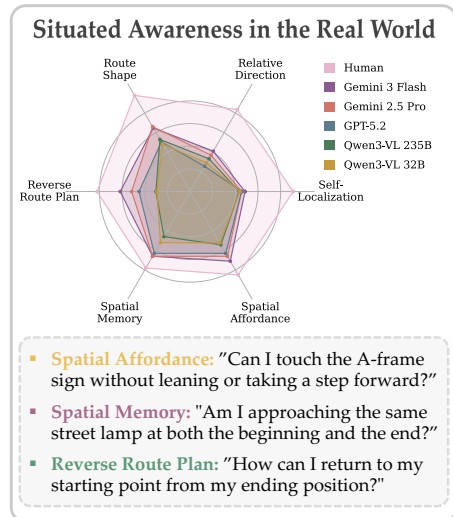

*Figure 1.* **(Left) Situated Awareness in the Real World.** A real-world example in which the observer walks along a straight trajectory while frequently rotating their head. The resulting egocentric video exhibits substantial camera orientation changes despite linear translational motion. Videos are recorded using Ray-Ban Meta (Gen 2) smart glasses. **(Right) Reasoning Task Performance.** Radar plot compares human performance with representative MFMs across six situated awareness tasks in SAW-BENCH.

## Abstract

A core aspect of human perception is *situated awareness*, the ability to relate ourselves to the surrounding physical environment and reason over possible actions in context. However, most existing benchmarks for multimodal foundation models (MFMs) emphasize **environment-centric** spatial relations (relations among objects in a scene), while largely overlooking **observer-centric** relationships that require reasoning relative to agent's viewpoint, pose, and motion. To bridge this gap, we introduce SAW-BENCH (**S**ituated **A**wareness in the Real **W**orld), a novel

benchmark for evaluating egocentric situated awareness using real-world videos. SAW-BENCH comprises 786 self-recorded videos captured with Ray-Ban Meta (Gen 2) smart glasses spanning diverse indoor and outdoor environments, and 2,071 *human-annotated* question-answer pairs. It probes a model's observer–environment understanding with *six* different awareness tasks. Our comprehensive evaluation reveals a human-model performance gap of 37.66%, even with the best-performing MFM, Gemini 3 Flash. Beyond this gap, our in-depth analysis uncovers several notable findings; for example, while models can exploit partial geometric cues in egocentric videos, they often fail to infer a coherent camera geometry, leading to systematic spatial reasoning errors. We position SAW-BENCH as a benchmark for situated spatial intelligence, moving beyond passive observation to understanding physically grounded, observer-centric dynamics.

*Equal contribution [1]University of California, Santa Barbara [2]Yale University [3]Stanford University [4]University of Maryland, College Park [5]Amazon [6]University of California, Merced. Correspondence to: Chuhan Li <chuhan_li@ucsb.edu>, Xin Eric Wang <ericxwang@ucsb.edu>.

*Proceedings of the 43rd International Conference on Machine Learning*, Seoul, South Korea. PMLR 306, 2026. Copyright 2026 by the author(s).

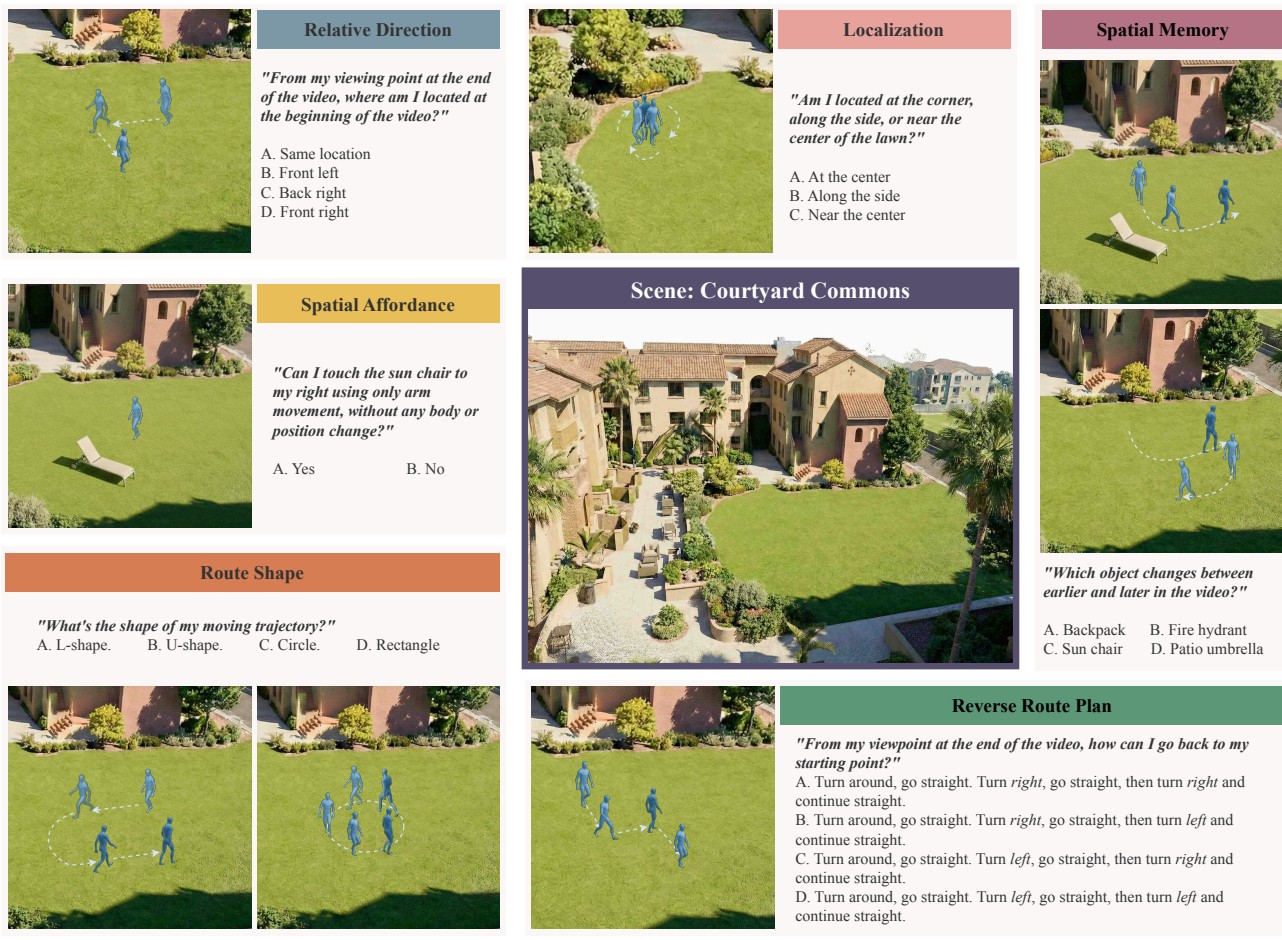

*Figure 2.* **Overview of SAW-BENCH.** We illustrate six representative tasks (§ 3.1) evaluating different aspects of situated awareness: Localization, Relative Direction, Route Shape, Reverse Route Plan, Spatial Memory, and Spatial Affordance. During data collection, human annotators follow pre-defined trajectories when recording egocentric videos (§ 3.2); these trajectories are visualized as purple dashed arrows. For all tasks, the model input consists solely of egocentric video without access to any bird's-eye or global scene representations; the visualizations shown here are provided for illustrative purposes only.

# 1. Introduction

Human perception of space is inherently *situated.* As people move through the world, they do not perceive scenes from a detached, global viewpoint; instead, they experience the environment relative to their own body and perspective. At any given moment, a person maintains an implicit sense of their location, orientation, and the "intentional arc" of their movements (Merleau-Ponty, 2013). Collectively, these **observer-centric** capabilities constitute **situated awareness** (Flach, 1995; Tversky, 2009; Sarter & Woods, 1991; Endsley, 1995), which operates continuously during everyday interaction and forms a foundational layer that supports more complex spatial intelligence.

Situated awareness is not only central to human perception, but also critical for autonomous systems operating in physical environments. In robotics, knowing *what* an object is is insufficient; an agent must also track precisely *where* it is

relative to its own body to effectively plan grasping and navigation. Similarly, in augmented and virtual reality (AR/VR), systems must continuously synchronize virtual content with the users' physical perspective to maintain immersion. In both domains, failure in this translation decouples the system's understanding from physical reality. Cognitive science studies suggest that such spatial intelligence relies on path integration, where local, situated updates are accumulated to a larger observer-aware map (McNaughton et al., 2006). Consequently, failures in situated awareness do not merely result in local errors; they create a cascading drift that significantly degrades the system's capability to understand space and time (Wolbers & Hegarty, 2010).

Yet, despite the importance of situated awareness, current evaluation landscape has largely overlooked these observer-centric capabilities. While there is growing interest in spatial reasoning for MFMs, existing benchmarks, such as VSI-Bench (Yang et al., 2025b) and MindCube (Wang et al.,

*Table 1.* **Summary of Visual-Spatial Reasoning Benchmarks.** We compare existing works from: [1]Egocentric setting, [2]Self-collected data, [3]Input modality, [4]Self-localization tasks, [5]Motion Understanding, [6]Trajectory Reasoning, [7]Spatial Memory, and [8]Action Feasibility.

| Benchmark | Ego. | Self-Col. | Mod. | Self-Loc. | Motion | Traj. | Mem. | Act. Feas. |
|---|---|---|---|---|---|---|---|---|
| Spatial-MM (Shiri et al., 2024) | ✓ | ✗ | Images | ✗ | ✗ | ✗ | ✗ | ✗ |
| ViewSpatial Bench (Li et al., 2025b) | ✓ | ✗ | Images | ✗ | ✗ | ✗ | ✗ | ✗ |
| Open3D-VQA (Zhang et al., 2025a) | ✗ | ✗ | Images | ✗ | ✗ | ✗ | ✗ | ✗ |
| SpatialBench (Xu et al., 2025) | ✓ | ✓ | Videos | ✗ | ✗ | ✗ | ✗ | ✗ |
| MindCube (Wang et al., 2026) | ✗ | ✓ | Images | ✗ | ✗ | ✗ | ✗ | ✗ |
| 3SDRBench (Ma et al., 2025) | ✗ | ✗ | Images | ✗ | ✗ | ✗ | ✗ | ✗ |
| All-Angles Bench (Yeh et al., 2026) | ✓ | ✗ | Images | ✗ | ✗ | ✗ | ✓ | ✗ |
| VSI-Super Recall (Yang et al., 2026b) | ✓ | ✗ | Videos | ✗ | ✗ | ✗ | ✓ | ✗ |
| VSI-Super Count (Yang et al., 2026b) | ✓ | ✗ | Videos | ✗ | ✗ | ✗ | ✓ | ✗ |
| OmniSpatial (Jia et al., 2026) | ✓ | ✗ | Images + Videos | ✗ | ✓ | ✗ | ✗ | ✓ |
| SpaCE-10 (Gong et al., 2026) | ✓ | ✗ | Images + Videos | ✗ | ✗ | ✓ | ✓ | ✗ |
| MomaGraph (Ju et al., 2026) | ✓ | ✗ | Images | ✗ | ✗ | ✓ | ✗ | ✓ |
| VLM4D (Zhou et al., 2025) | ✓ | ✗ | Videos | ✗ | ✓ | ✗ | ✗ | ✗ |
| MMSI-Bench (Yang et al., 2026a) | ✓ | ✗ | Images | ✗ | ✓ | ✗ | ✗ | ✗ |
| VSI-Bench (Yang et al., 2025b) | ✓ | ✗ | Videos | ✗ | ✓ | ✗ | ✓ | ✗ |
| MMSI-Video-Bench (Lin et al., 2025) | ✓ | ✓ | Videos | ✗ | ✓ | ✗ | ✓ | ✗ |
| **SAW-BENCH** (Ours) | ✓ | ✓ | Videos | ✓ | ✓ | ✓ | ✓ | ✓ |

2026), predominantly focus on observer-independent tasks. These benchmarks emphasize object-object interaction, discrete mental simulation, and distance estimation – skills that define spatial reasoning from a detached, static, and third-person perspective. As a result, models are evaluated under the assumption that they are passive spectators rather than active embodied agents with their own viewpoint, motion, and position. This leaves observer-centric situated awareness, the ability to understand space relative to the observer and how it evolves over time, largely unexplored.

To bridge this gap, we introduce SAW-BENCH, **S**ituated **A**wareness in the Real **W**orld, a novel video understanding benchmark designed to assess MFMs' situated awareness capabilities. SAW-BENCH comprises 2,071 carefully curated, human-annotated multiple-choice question-answer pairs spanning six distinct awareness tasks (localization, relative direction, route shape, reverse route plan, spatial memory, and spatial affordance), applied to 786 egocentric videos encompassing both indoor and outdoor environments (§3.2). Videos in SAW-BENCH are all self-recorded using Ray-Ban Meta (Gen 2) glasses. The question-answer pairs are carefully designed to ensure that models need to reason about the observer itself and the environment, making situated awareness essential for solving our tasks (§3.1).

We conduct a comprehensive evaluation of 17 open-source and 6 proprietary MFMs. The best-performing model, Gemini 3 Flash, yields a performance of 53.89% on SAW-BENCH, significantly below that of human performance (91.55%). Beyond this overall performance gap, we identify four systematic patterns in model behavior (§5): (1) models often conflate egocentric camera rotation with translational movement; (2) model accuracy degrades significantly as trajectory complexity increases; (3) models exhibit a mismatch between view-level internal memory and persistent world-state memory; and (4) environment openness alone is an insufficient proxy for spatial reasoning difficulty.

## 2. Related Work

**General video understanding benchmarks.** Video understanding has become a central capability in MFMs, serving as a key testbed for evaluating their ability to perceive, reason, and integrate visual and linguistic information over time. Recent benchmarks emphasize more complex forms of reasoning such as long-form video understanding (Chandrasegaran et al., 2024; Wu et al., 2024; Chen et al., 2025; Wang et al., 2025); visual-temporal reasoning (Liu et al., 2024; Shangguan et al., 2025; Xue et al., 2025); domain-specific reasoning (He et al., 2025; Zhao et al., 2025; Deng et al., 2025); and comprehensive video understanding (Ning et al., 2026; Li et al., 2024; Fu et al., 2025). Despite this growing complexity, these benchmarks remain largely allocentric, evaluating models as passive observers of scene-level events. Consequently, observer-centric spatial reasoning, understanding one's own position and relationship to the environment, remains largely unexamined, motivating the curation of SAW-BENCH.

**3D spatial intelligence.** Research in 3D spatial intelligence has predominantly focused on reasoning over explicit, reconstructed geometric representations such as point clouds and meshes (Jain et al., 2022; Hsu et al., 2023; Huang et al., 2022; Bakr et al., 2024). Early datasets like ReferIt3D (Achlioptas et al., 2020) and ScanRefer (Chen et al., 2020) evaluate the capability to ground natural language into specific 3D coordinates, while more recent datasets like 3D-VisTA (Zhu et al., 2023), 3D-GRAND (Yang et al., 2025a), and EmbodiedScan (Wang et al., 2024a) assess more complex grounding and reasoning over holistic 3D scenes. MSQA (Linghu et al., 2024) further evaluates object existence, spatial relationships, and next-step navigation, though it remains grounded in structured 3D environments. Among these datasets, SQA3D (Ma et al., 2023) is the most relevant to our work, as it explicitly introduces situated

questions anchored to a specific position and orientation within a 3D scene. Although SQA3D supports video-based evaluation using egocentric videos from ScanNet (Dai et al., 2017), these were captured for 3D reconstruction and scene coverage rather than natural human movement, and the "situation" is specified as a static pose with a fixed position and facing direction rather than context that emerges from an ongoing trajectory. In contrast, SAW-BENCH defines situations dynamically through the observer's trajectory and evaluates observer-centric capabilities, spanning both navigation and non-navigation aspects such as spatial memory and affordance, directly from raw egocentric video. Unlike reconstructed 3D scenes, which are costly to obtain and difficult to scale, egocentric video is a lower-friction input that requires neither 3D capture nor externally mounted cameras, better reflecting practical embodied perception.

**Visual-spatial intelligence benchmarks.** In the visual domain, existing benchmarks often focus on high-level semantic or commonsense reasoning, while overlooking fine-grained and precise spatial intelligence. Evaluations that do focus on spatial awareness largely frame spatial reasoning through observer-independent tasks, overlooking the inherently observer-centric nature of embodied tasks. Benchmarks such as VSI-Bench (Yang et al., 2025b), VSI-Super (Yang et al., 2026b), and SpaCE-10 (Gong et al., 2026) emphasize object-object interaction, counting, and distance estimation, which treat models as a detached, external observer. Similarly, MindCube (Wang et al., 2026), MomaGraph (Ju et al., 2026) and All-Angles Bench (Yeh et al., 2026) assess spatial reasoning using discrete multi-view inference. While they evaluate models' ability to integrate spatial information across viewpoints, they typically assume the observer's state is given or static. As a result, models' capability for **situated awareness**, the continuous real-time update from observer's own pose and perspective relative to the environment, largely unexplored. A more comprehensive comparison is provided in Table 1.

## 3. Situated Awareness Benchmark

### 3.1. Situated Awareness Tasks in SAW-BENCH

While cognitive science does not prescribe a canonical task taxonomy for situated awareness, prior work across navigation (Tversky, 1993; Burgess, 2006), spatial updating (Franklin et al., 1992; Michon & Denis, 2001), spatial working memory (Luck & Vogel, 1997; Simons & Levin, 1997), and affordance (Gibson & Walk, 1960; Gibson, 2014) has studied these abilities as separable components, each capturing a distinct aspect of how observers relate themselves to the environment. Motivated by this decomposition, we introduce six situated awareness tasks over 2,071 human-annotated question-answer pairs derived from 786 videos,

each of which requires models to understand and reason over the relationship between itself and the environment:

- **Self-Localization** (9.66%): infer the observer's position within the environment from an egocentric viewpoint;
- **Relative Direction** (40.27%): reason about the observer's relative position across time by relating starting and ending viewpoints;
- **Route Shape** (26.36%): characterize the geometric shape of the observer's movement trajectory;
- **Reverse Route Plan** (11.06%): infer a sequence of movements that returns the observer to the starting location;
- **Spatial Memory** (4.83%): reason about changes in the environment by comparing spatial information across time;
- **Spatial Affordance** (7.82%): determine whether a specific action is feasible under physical constraints from the observer's viewpoint.

### 3.2. Video Collection

Egocentric video is a natural sensing modality for studying situated awareness, as it captures the environment from the observer's own viewpoint and preserves the observer-centric structure of spatial perception during real-world interaction. Unlike third-person footage, egocentric video directly encodes where objects appear relative to the camera wearer, how the field of view evolves with head and body movement, and how visibility of the environment changes over time.

To reflect this setting, all videos in SAW-BENCH are recorded from an egocentric perspective using Ray-Ban Meta (Gen 2) smart glasses worn by human participants. Most videos are captured as single, continuous clips without interruption. For tasks involving **Spatial Memory**, we apply limited post-processing by concatenating two short clips recorded in the same physical scene: one before and one after a controlled modification of the environment. No other temporal reordering or editing is performed. Audio is excluded from all videos to ensure that all reasoning is grounded solely in visual information.

Our video collection process spans a diverse set of real-world environments, including 10 outdoor scenes (*e.g.,* courtyards, parking lots, lawns, and plazas) and 5 indoor scenes (*e.g.,* lecture halls, classrooms, recreation rooms, and household environments). Within each scene, we collect approximately 40-60 distinct videos to support tasks that benefit from dense coverage of a fixed environment, such as **Self-Localization** and **Route Shape**. For tasks that are more difficult to scale within a limited set of scenes, particularly **Spatial Memory** and **Spatial Affordance**, we additionally collect a set of videos across a broader range of environments outside these core scenes. This supplemental collection prioritizes diversity over dense coverage,

*Figure 3.* **Benchmark Curation Pipeline.** We first pre-define 46 camera trajectories and annotate their metadata. Human video collectors then record egocentric videos by following these trajectories in selected scenes. Low-quality recordings are filtered and re-captured to ensure consistent video quality.

enabling evaluation of memory and action feasibility across varied layouts and physical constraints without requiring exhaustive sampling of each scene.

During video collection, participants followed a lightweight recording protocol, consisting of high-level guidelines intended to ensure consistency across scenes while preserving natural behavior. For tasks involving Self-Localization, participants were instructed to record videos from a set of predefined reference locations (*e.g.,* corners, side, or center) to ensure coverage of diverse viewpoints. Beyond these coverage requirements, the protocol did not prescribe specific paths, motions, or camera poses. Instead, participants were instructed to follow coarse trajectory shapes (*e.g.,* straight line, zigzag, or two consecutive turns), while retaining flexibility in how these shapes were executed within each environment. Recording protocols are provided in §D.1.

### 3.3. Question-Answer Annotation and Quality Check

**Question-answer annotation.** Question-answer (QA) pairs in SAW-BENCH are annotated based on the predefined recording protocol and the known trajectory of each video; the full list of questions is provided in §D. All QA pairs are annotated by the same human participants who recorded the videos, who followed predefined reference locations and coarse trajectory shapes during data collection. As a result, annotation is restricted to these predefined configurations rather than open-ended interpretation of the video content. This design allows questions to target well-defined aspects of agent-centric situated awareness while minimizing ambiguity in ground-truth answers.

**Quality control.** We perform quality checks at both the video and annotation levels. For video quality control, each recorded video is manually reviewed by human reviewers. Videos exhibiting issues such as rapid head motion, poor visibility of key objects, or other factors that could impair spatial reasoning were discarded and re-filmed following the same recording protocol. To ensure high-quality QA annotation, each QA pair in the protocol is independently annotated by two annotators. We report the inter-annotator agreement score in §D.2. Disagreements were resolved

through a final review process following the same annotation guidelines. This rigorous approach ensures consistency and accuracy across all annotated QA pairs.

## 4. Experiments

### 4.1. Evaluation Protocol

**Models.** We evaluate a diverse set of general-purpose MFMs in a zero-shot setting. For *open-source* MFMs, we test: Qwen2.5-VL (Bai et al., 2025b), Qwen3-VL (Bai et al., 2025a), InternVL 2 (Chen et al., 2024), InternVL 3 (Zhu et al., 2025), LLaVA-NeXT-Video (Zhang et al., 2025b), and LLaVA-OneVision (Li et al., 2025a). We also evaluate the following *proprietary* MFMs: GPT (Singh et al., 2026) and Gemini (Gemini Team, 2025; Comanici et al., 2025). We provide full model evaluation configuration in Table 4.

**Baselines.** We include five baselines: (1) *chance level (random)*, which is defined as the expected accuracy under uniform random answer selection; (2) *chance level (frequent)*, which is defined as the accuracy achieved by always selecting the most frequent answer; (3) *blind LLM*, which answers the multiple-choice questions without access to any visual information from the video. We use GPT-5.2 as our blind evaluation for this baseline. Details are provided in §C.1; (4) *socratic models* (Zeng et al., 2023), which generate a single holistic caption for each video using a video captioner and use this caption as a language-based representation of the video for downstream question answering. In evaluation, the model is provided with the question and the caption only. We use GPT-5.2 for both video captioning and question answering. Details are provided in §C.2; (5) *human level*, which is measured based on the answers given by two graduate students, who have access to the full videos and no time constraints. Details are provided in §C.3.

**Accuracy evaluation.** We use accuracy as the primary metric to evaluate model performance on SAW-BENCH. Following recent benchmarks for foundation model evaluation (Wang et al., 2024b; Shangguan et al., 2025; Zhao et al., 2025), we first apply a regular-expression–based parser to extract the predicted answer from each model's raw re-

*Table 2.* **Evaluation Results on SAW-BENCH.** Unless otherwise specified, all models process videos at 2 fps (frames per second). Frame level sensitivity analyses are provided in §F. **Bold** and underlined numbers indicate the best and second-best performance in each category, respectively. Model configurations are provided in Table 4. \*: Human baseline details are provided in §C.3. §: Models do not support fps-based sampling and process a fixed total of 32 frames per video. ¶: 8 frames per video sampling due to compute limitations.

| Model | All | Self-Localization | Relative Direction | Route Shape | Reverse Route Plan | Spatial Memory | Spatial Affordance |
|---|---|---|---|---|---|---|---|
| **Baselines** | | | | | | | |
| **Human Level** [*] | **91.55** | **94.00** | **89.39** | **97.62** | **93.01** | **88.50** | **79.01** |
| Chance Level (Random) | 27.49 | 34.00 | 25.90 | 21.43 | 27.51 | 28.00 | 56.17 |
| Chance Level (Frequent) | 29.55 | 38.00 | 25.90 | 27.11 | 27.51 | 27.00 | 50.62 |
| Blind LLM (GPT-5.2) | 31.34 | 38.00 | 23.02 | 36.63 | 24.02 | 38.00 | 54.32 |
| Socratic Model (GPT-5.2) | 31.34 | 40.50 | 20.62 | 41.58 | 24.02 | 32.00 | 50.62 |
| **Proprietary Multimodal Foundation Models** | | | | | | | |
| Gemini 3 Flash | **53.89** | 48.50 | **41.13** | 64.84 | **61.57** | **66.00** | **70.99** |
| Gemini 2.5 Pro | 50.80 | 45.50 | 37.05 | **66.12** | 51.53 | **66.00** | 66.05 |
| Gemini 3 Pro | 45.97 | **50.00** | 38.61 | 52.01 | 36.24 | 63.00 | 61.73 |
| GPT-5.2 | 41.04 | 45.50 | 25.78 | 50.55 | 44.98 | 63.00 | 62.96 |
| Gemini 2.5 Flash | 39.79 | 44.00 | 25.30 | 57.33 | 37.99 | 49.00 | 46.91 |
| GPT-5 Mini | 33.80 | 43.50 | 27.46 | 36.08 | 22.27 | 56.00 | 49.38 |
| **Open-Source Multimodal Foundation Models** | | | | | | | |
| Qwen3-VL 235B-A22B | **41.40** | 43.50 | **33.41** | **53.11** | **30.13** | 46.00 | 54.32 |
| Qwen3-VL 32B | 38.58 | 44.00 | 29.14 | 48.35 | 29.26 | 52.00 | 52.47 |
| Qwen3-VL 30B-A3B | 36.55 | 39.00 | 29.62 | 43.04 | 27.07 | **54.00** | 50.00 |
| Qwen2.5-VL 32B | 36.46 | **53.00** | 28.06 | 41.03 | 24.89 | 45.00 | 54.94 |
| Qwen2.5-VL 72B | 36.17 | 51.50 | 26.74 | 41.76 | 25.33 | 45.00 | **56.79** |
| Qwen3-VL 8B | 36.12 | 40.00 | 27.82 | 46.70 | 23.58 | 48.00 | 48.77 |
| LLaVA OneVision 72B[§] | 33.70 | 39.00 | 22.30 | 46.15 | 24.45 | 41.00 | 52.47 |
| InternVL3 8B[§] | 33.70 | 43.50 | 26.86 | 36.45 | 27.95 | 46.00 | 48.15 |
| LLaVA-Video 72B[§] | 32.98 | 32.50 | 23.86 | 43.04 | 24.45 | 41.00 | 53.70 |
| InternVL3 14B[§] | 32.69 | 49.00 | 17.27 | 45.05 | 24.02 | 54.00 | 49.38 |
| Qwen2.5-VL 7B | 31.48 | 38.50 | 19.06 | 43.59 | 26.20 | 38.00 | 49.38 |
| LLaVA-NeXT-Video 32B[§] | 31.24 | 41.00 | 24.46 | 35.35 | 22.27 | 34.00 | 51.23 |
| LLaVA-Video 7B[§] | 30.81 | 41.00 | 25.06 | 32.78 | 24.45 | 32.00 | 49.38 |
| InternVL2 40B[¶] | 30.13 | 45.00 | 17.75 | 38.28 | 24.89 | 32.00 | 54.32 |
| InternVL2 8B[§] | 29.84 | 43.00 | 14.99 | 41.94 | 24.89 | 40.00 | 50.00 |
| LLaVA OneVision 7B[§] | 29.45 | 34.50 | 20.26 | 34.80 | 25.33 | 44.00 | 49.38 |
| InternVL3 38B[¶] | 27.71 | 35.50 | 23.50 | 37.55 | 24.45 | 46.00 | 51.23 |

sponse. If the parser fails, we additionally use GPT-4o-mini to extract the answer from the raw output. Prompt used for answer extraction is provided in §B.3.

### 4.2. Main Results

We provide quantitative results on SAW-BENCH for all models in Table 2. To better understand where models fail, we select a set of representative models (Gemini Team, 2025; Comanici et al., 2025; Singh et al., 2026; Bai et al., 2025a;b) and present examples of failure cases in §G.1, §G.2, §G.3, §G.4, §G.5, and §G.6 for each of the reasoning types in SAW-BENCH. Overall, our results reveal that situated awareness remains a significant challenge for current models, with even the best-performing model falling well short of human performance. Below, we summarize key findings from the main results.

**Widespread difficulty in situated awareness.** Our evaluation reveals that situated awareness remains a fundamental challenge for current MFMs. Even the top-performing model, Gemini 3 Flash, achieves only 53.89% accuracy overall, leaving a 37.66% performance gap compared to human accuracy of 91.55%. Notably, humans perform remarkably

well on **Self-Localization** and **Route Shape** tasks, suggesting that these categories are naturally intuitive to observers. Interestingly, the performance gap between human and top-performing MFMs is much smaller on **Spatial Memory** and **Spatial Affordance**, highlighting that current models may be relatively strong on tasks that rely more heavily on spatial memorization and depth cues.

**Proprietary *vs.* open-source model.** In general, proprietary MFMs outperform open-source MFMs, with the largest performance gap appearing on **Reverse Route Plan**, a task that requires sustained reasoning over egocentric trajectories and explicit tracking of intermediate movements. We provide a qualitative analysis on model responses in Figure 4. In these cases, open-source models tend to rely on shortcuts cues from some "key frames," typically the first and the last frame of the video; whereas proprietary models more consistently maintain a coherent observer-centric representation across the full extent of the video.

**Blind LLM *vs.* socratic model.** The blind LLM achieves an overall accuracy of 31.34%, which is only marginally above chance level baselines, indicating that effective per-

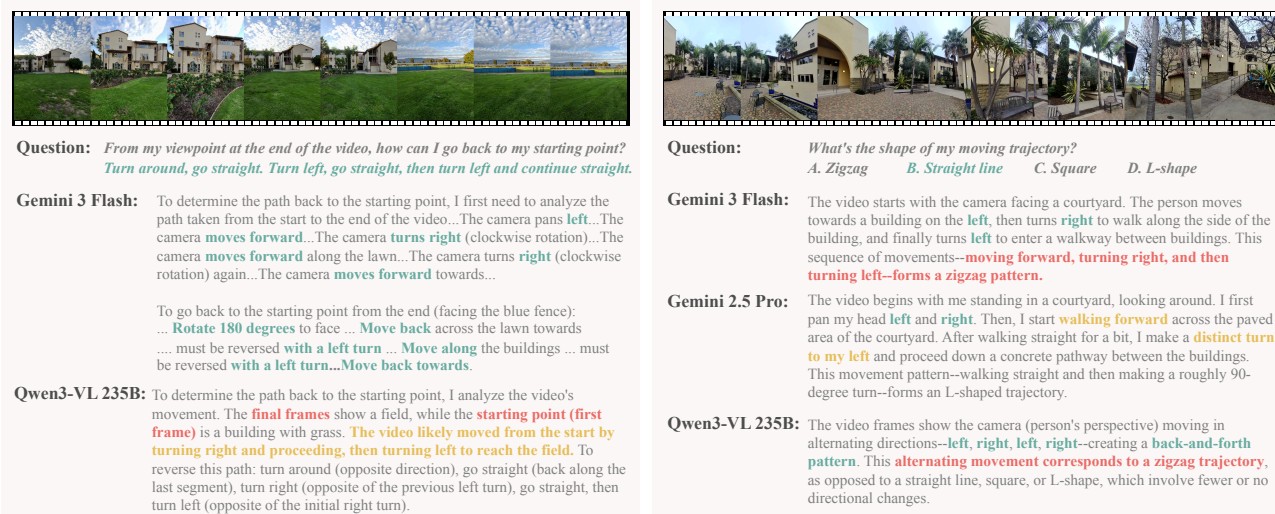

**Question:** *From my viewpoint at the end of the video, how can I go back to my starting point? Turn around, go straight. Turn left, go straight, then turn left and continue straight.*

**Gemini 3 Flash:** To determine the path back to the starting point, I first need to analyze the path taken from the start to the end of the video...The camera pans **left**...The camera **moves forward**...The camera **turns right** (clockwise rotation)...The camera **moves forward** along the lawn...The camera turns **right** (clockwise rotation) again...The camera **moves forward** towards...

To go back to the starting point from the end (facing the blue fence): ... **Rotate 180 degrees** to face ... **Move back** across the lawn towards .... must be reversed **with a left turn** ... **Move along** the buildings ... must be reversed **with a left turn**...**Move back towards**.

**Qwen3-VL 235B:** To determine the path back to the starting point, I analyze the video's movement. The **final frames** show a field, while the **starting point (first frame)** is a building with grass. **The video likely moved from the start by turning right and proceeding, then turning left to reach the field.** To reverse this path: turn around (opposite direction), go straight (back along the last segment), turn left (opposite of the previous left turn), go straight, then turn left (opposite of the initial right turn).

**Question:** *What's the shape of my moving trajectory?*
*A. Zigzag   B. Straight line   C. Square   D. L-shape*

**Gemini 3 Flash:** The video starts with the camera facing a courtyard. The person moves towards a building on the **left**, then turns **right** to walk along the side of the building, and finally turns **left** to enter a walkway between buildings. This sequence of movements--**moving forward, turning right, and then turning left--forms a zigzag pattern.**

**Gemini 2.5 Pro:** The video begins with me standing in a courtyard, looking around. I first pan my head **left** and **right**. Then, I start **walking forward** across the paved area of the courtyard. After walking straight for a bit, I make a **distinct turn to my left** and proceed down a concrete pathway between the buildings. This movement pattern--walking straight and then making a roughly 90-degree turn--forms an L-shaped trajectory.

**Qwen3-VL 235B:** The video frames show the camera (person's perspective) moving in alternating directions--**left**, **right**, **left**, **right**--creating a **back-and-forth pattern**. This **alternating movement corresponds to a zigzag trajectory**, as opposed to a straight line, square, or L-shape, which involve fewer or no directional changes.

*Figure 4.* **Error Case Analysis.** (**Left**) Reverse Route Plan: Gemini 3 Flash successfully reconstructs the return path by systematically inverting the actions from the forward pass. In contrast, Qwen3-VL 235B attempts to exploit a shortcut between the first and last frames, thereby neglecting the transitive dynamics and spatial transformations occurring throughout the frame sequence. (**Right**) Route Shape: While both Gemini 3 Flash and Qwen3-VL 235B effectively identify camera rotations, they falsely integrate these rotational pans into the observer's physical movement trajectory, leading to incorrect shape understanding.

formance on SAW-Bench requires access to visual information. Compared to Blind LLM, Socratic model does not yield significant performance gains, achieving the same overall accuracy of 31.34%. Although Socratic model has indirect access to visual content through video captioning, reducing egocentric video into a static language-based representation discards critical observer-centric cues such as viewpoint changes, orientation, and temporal structure. Notably, the Socratic model exhibits slightly better performance on Route Shape than the Blind LLM, suggesting that captions can convey coarse trajectory-level information. However, this limited improvement does not extend to other tasks. As a result, high-level semantic summaries alone are insufficient for situated awareness.

## 5. Analysis

To better understand when and why models fail at situated awareness, we analyze representative error patterns that reflect core components of observer-centric reasoning.

**Camera rotation as a source of trajectory errors.** We identify a systematic failure mode in Route Shape occurring when changes in camera rotation are decoupled from the observer's translational movement. To isolate this effect, we compare three controlled scenarios: (1) a straight path with stable head orientation (Figure 5 **Left**); (2) the same straight path with frequent head rotations (Figure 5 **Middle**); and (3) a true zigzag trajectory (Figure 5 **Right**).

Despite identical translational motion in cases (1) and (2),

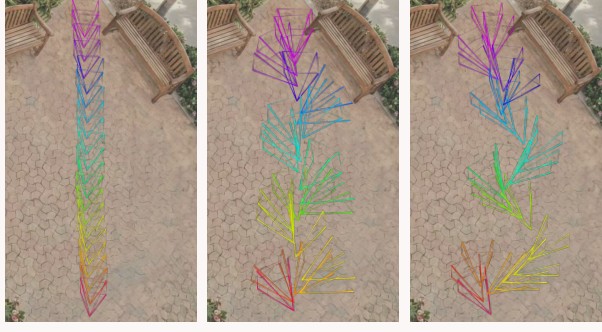

*Figure 5.* **Camera Rotation and Observer's Trajectory.** Visualization of three controlled scenarios used to isolate the impact of head rotation on Route Shape. (**Left**) a straight path with steady head orientation; (**Middle**) the same straight path with frequent left-and-right head rotations; and (**Right**) a true zigzag trajectory.

even top-performing models frequently misclassify case (2) as a zigzag trajectory: Gemini 3 Flash does so in 60.0% of instances, while Qwen3-VL 235B misclassifies 53.3% of cases. As illustrated in Figure 4 **Right**, models justify these predictions by erroneously attributing camera orientation shifts to physical body displacement. This failure highlights a fundamental limitation in current MFMs: the inability to maintain a robust observer-centric coordinate system that distinguishes egocentric rotational pans from global positional updates.

*Table 3.* **Accuracy (%) on** Relative Direction **Tasks Stratified by the Number of Turns**. Performance for most models degrades significantly as geometric complexity increases.

| Models | Straight | Single Turn | Two Turns |
|---|---|---|---|
| **Human** | **100.00** | **96.67** (-3.33%) | **90.00** (-10.00%) |
| Gemini 3 Flash | 73.33 | 70.69 (-3.60%) | 40.61 (-44.63%) |
| Gemini 3 Pro | 63.33 | 56.90 (-10.16%) | 36.46 (-42.44%) |
| Gemini 2.5 Pro | 73.33 | 55.17 (-24.76%) | 33.41 (-54.44%) |
| GPT-5.2 | 30.00 | 39.66 (+32.20%) | 22.49 (-25.03%) |
| Qwen3-VL 235B | 90.00 | 8.62 (-90.42%) | 27.85 (-69.06%) |
| Qwen3-VL 32B | 80.00 | 12.07 (-84.91%) | 21.83 (-72.71%) |

> **Finding 1.** MFMs often conflate **egocentric camera rotation** with **translational movement**.

**Trajectory complexity and error accumulation.** Spatial updating is an inherently accumulative process, where errors in estimating egocentric motion compound as the observer moves through an environment (McNaughton et al., 2006; Wolbers & Hegarty, 2010; Stangl et al., 2020). In human navigation, this integration is highly sensitive to "noise" introduced by changes in orientation (Cherep et al., 2020).

To investigate whether MFMs exhibit a similar sensitivity to trajectory complexity, we stratify results on the Relative Direction task by geometric complexity: (1) **Straight** (pure translation), (2) **Single Turn** (one rotational update), and (3) **Two Turns** (multiple rotational updates). As shown in Table 3, increasing geometric complexity is associated with a substantial accuracy degradation for most models, particularly when trajectories involve multiple orientation changes. When quantified using relative performance drop with respect to straight trajectories, MFMs often exhibit significant degradation under multi-turn conditions, while human performance remains largely stable. This human-model gap suggests that current MFMs struggle to reliably integrate successive egocentric orientation changes over time, resulting in compounding errors as trajectories move away from simple translational motion.

> **Finding 2.** Model accuracy **degrades** significantly as trajectory complexity increases.

**Failure to maintain persistent object memory.** A recurring failure mode across Spatial Memory tasks arises from models' difficulty in maintaining object persistence across egocentric motion. Although models often provide accurate descriptions of what is visible in individual frames or short temporal windows, they fail to reason about objects that leave the camera's field of view. As shown in Figure 6, models tend to infer that objects are absent in earlier frames simply because they are not visible, incorrectly treating first observation as object appearance rather than recognizing

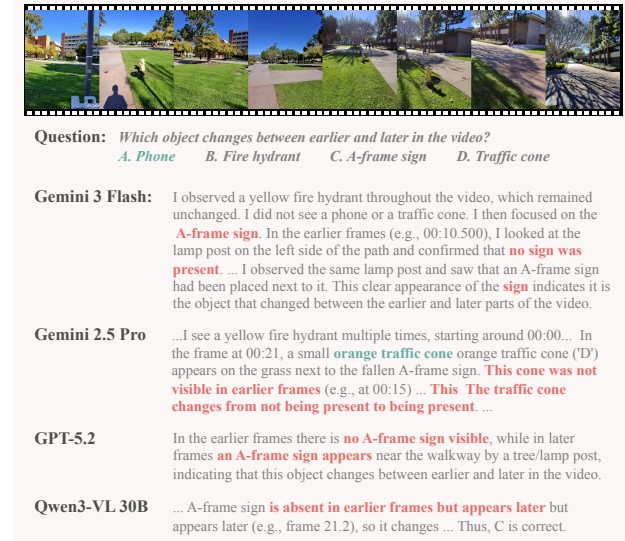

**Question:** *Which object changes between earlier and later in the video?*
*A. Phone    B. Fire hydrant    C. A-frame sign    D. Traffic cone*

**Gemini 3 Flash:** I observed a yellow fire hydrant throughout the video, which remained unchanged. I did not see a phone or a traffic cone. I then focused on the A-frame sign. In the earlier frames (e.g., 00:10.500), I looked at the lamp post on the left side of the path and confirmed that no sign was present. ... I observed the same lamp post and saw that an A-frame sign had been placed next to it. This clear appearance of the sign indicates it is the object that changed between the earlier and later parts of the video.

**Gemini 2.5 Pro** ...I see a yellow fire hydrant multiple times, starting around 00:00... In the frame at 00:21, a small orange traffic cone orange traffic cone ('D') appears on the grass next to the fallen A-frame sign. This cone was not visible in earlier frames (e.g., at 00:15) ... This The traffic cone changes from not being present to being present. ...

**GPT-5.2** In the earlier frames there is no A-frame sign visible, while in later frames an A-frame sign appears near the walkway by a tree/lamp post, indicating that this object changes between earlier and later in the video.

**Qwen3-VL 30B** ... A-frame sign is absent in earlier frames but appears later but appears later (e.g., frame 21.2), so it changes ... Thus, C is correct.

*Figure 6.* **Model Responses in** Spatial Memory. Across multiple models, non-visibility is incorrectly treated as non-existence: objects that exit the camera's field of view are inferred to have disappeared or changed, revealing a gap between *what is seen* and *what exists*.

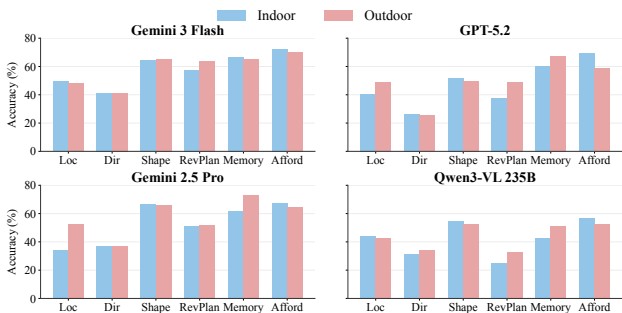

*Figure 7.* **Indoor** *vs.* **Outdoor Performance**. Comparison of zero-shot accuracy across six situated awareness tasks for Gemini 3 Flash, Gemini 2.5 Pro, GPT-5.2, and Qwen3-VL 235B.

that the object may have existed outside the field of view. These errors suggest that current MFMs rely primarily on view-dependent evidence, rather than maintaining a persistent world-state representation over time.

> **Finding 3.** **Persistent tracking** of objects across frames remains an open challenge across models.

**Effect of openness on situated awareness.** Figure 7 summarizes model performance across indoor and outdoor environments. Contrary to the intuition that larger and more dynamic outdoor environments may increase spatial reasoning difficulty, no consistent performance degradation is observed in outdoor scenes. Across the four selected models, outdoor performance is often comparable to, and in

several cases higher than, indoor performance. On average, the indoor–outdoor performance gap remains small.

These results suggest that environment scale alone does not determine spatial reasoning difficulty. While outdoor scenes typically span larger spatial extents, they often contain fewer objects and exhibit less structural clutter than indoor environments, which may reduce relational ambiguity. As a result, spatial reasoning difficulty is not monotonically correlated with scene size or openness. Instead, indoor environments can pose equally, if not more, complex spatial challenges due to higher object density and more intricate layout structures.

> **Finding 4.** Environment openness alone is an **insufficient proxy** for spatial reasoning difficulty.

## 6. Conclusion

*Situated awareness*, the capacity to relate oneself to the surrounding physical environment and reason over possible actions in context, is a foundational component of human spatial cognition. Yet, it has remained largely unexplored in current multimodal evaluation frameworks, which often treat models as passive observers rather than embodied agents with their own viewpoint, motion, and position. To bridge this gap, we introduced SAW-BENCH, a benchmark of 2,071 human-annotated question–answer pairs over 786 egocentric videos, designed to probe six distinct facets of observer-centric spatial understanding. Our evaluation of 23 multimodal foundation models reveals a substantial human–model performance gap: even the best-performing model, Gemini 3 Flash, achieves 53.89% accuracy, compared with 91.55% for humans. Beyond this performance gap, our analysis shows that current models conflate egocentric camera rotation with translational movement, degrade as trajectory complexity increases, and fail to maintain persistent object representations beyond the current field of view. These failures suggest that current multimodal models do not yet reliably accumulate and update observer-centric spatial state over time. We hope SAW-BENCH can provide a useful diagnostic testbed for developing multimodal systems that reason not only about what is visible, but also about where they are, how they have moved, and how they can act within dynamic physical environments.

## Impact Statement

This work introduces SAW-BENCH, a benchmark for evaluating observer-centric situated awareness in MFMs using egocentric videos. SAW-BENCH provides a diagnostic tool to measure spatial understanding capabilities that are currently underrepresented in existing evaluation frameworks.

Potential impacts include improved reliability of AI systems deployed in robotics, augmented and virtual reality (AR/VR), and assistive technologies, where understanding spatial relationships from embodied agent's or human wearer's perspective is critical for safe and effective operation. While our benchmark does not introduce direct pathways to harm, we acknowledge potential downstream misuse, as models that perform well on these tasks could be integrated into applications that may be deployed in harmful ways. We encourage future work to study and adopt responsible deployment practices for systems built on or evaluated with our benchmark.

## Acknowledgment

We thank Ziyao Shangguan, Qianqi Yan, Gurusha Juneja, Adina Roskies, Mable Zhou, and Mary Hegarty for their helpful discussion and feedback. This project is partially sponsored by an Amazon gift award.

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

# Contents

# A. Experiment Setup

## A.1. Model Configuration

*Table 4.* **Model Configurations for Evaluation.** Unset values indicate that their default values are being used. Configurations are based on official model repositories where available. **Temp.**: temperature. [‡]: GPT-5 mini only accepts a temperature value of 1.0. [*]: GPT-5.2 does not support adjusting the temperature or top-p parameters. [†]: The thinking_level parameter for Gemini 3 Pro was set to low. When configured to dynamic, Gemini 3 Pro consistently generated reasoning traces that exceeded the output context window, resulting in incomplete or Null responses. [§]: Models do not support fps-based sampling and process a fixed total of 32 frames per video. [¶]: 8 frames per video sampling due to compute limitations.

| Model | API Checkpoint / HF Checkpoint | Max New Tokens | Temp. | Top-P | Top-K | Video Sampling | Thinking Level |
|---|---|---|---|---|---|---|---|
| **Proprietary Multimodal Foundation Models** | | | | | | | |
| Gemini 3 Pro[†] | gemini-3-pro-preview | 16384 | 0.0 | 1.0 | 1.0 | 2 fps | low[†] |
| Gemini 3 Flash | gemini-3-flash-preview | 16384 | 0.0 | 1.0 | 1.0 | 2 fps | |
| Gemini 2.5 Pro | gemini-2.5-pro | 16384 | 0.0 | 1.0 | 1.0 | 2 fps | |
| Gemini 2.5 Flash | gemini-2.5-flash | 16384 | 0.0 | 1.0 | 1.0 | 2 fps | |
| GPT-5.2[*] | gpt-5.2-2025-12-11 | 16384 | | | | 2 fps | medium |
| GPT-5 mini[‡] | gpt-5-mini-2025-08-07 | 16384 | 1.0 | 1.0 | | 2 fps | medium |
| **Open-Source Multimodal Foundation Models** | | | | | | | |
| Qwen3-VL 235B | qwen3-vl-235b-a22b-thinking | 16384 | 0.0 | 1.0 | | 2 fps | medium |
| Qwen3-VL 32B | qwen3-vl-32b-thinking | 16384 | 0.0 | 1.0 | | 2 fps | medium |
| Qwen3-VL 30B | qwen3-vl-30b-a3b-thinking | 16384 | 0.0 | 1.0 | | 2 fps | medium |
| Qwen3-VL 8B | qwen3-vl-8b-thinking | 16384 | 0.0 | 1.0 | | 2 fps | medium |
| Qwen2.5-VL 72B | qwen2.5-vl-72b-instruct | 8192 | 0.0 | 1.0 | | 2 fps | |
| Qwen2.5-VL 32B | qwen2.5-vl-32b-instruct | 8192 | 0.0 | 1.0 | | 2 fps | |
| Qwen2.5-VL 7B | qwen2.5-vl-7b-instruct | 8192 | 0.0 | 1.0 | | 2 fps | |
| LLaVA-NeXT-Video 32B[§] | LLaVA-NeXT-Video-32B-Qwen | 16384 | 0.0 | 1.0 | | 32 frames[§] | |
| LLaVA-Video 7B[§] | LLaVA-Video-7B-Qwen2 | 16384 | 0.0 | 1.0 | | 32 frames[§] | |
| LLaVA-Video 72B[§] | LLaVA-Video-72B-Qwen2 | 16384 | 0.0 | 1.0 | | 32 frames[§] | |
| LLaVA OneVision 72B[§] | llava-onevision-qwen2-72b-ov | 16384 | 0.0 | 1.0 | | 32 frames[§] | |
| LLaVA OneVision 7B[§] | llava-onevision-qwen2-7b-ov | 16384 | 0.0 | 1.0 | | 32 frames[§] | |
| InternVL3 38B[¶] | InternVL3-38B | 512 | 0.0 | 1.0 | | 8 frames[¶] | |
| InternVL3 14B[§] | InternVL3-14B | 512 | 0.0 | 1.0 | | 32 frames[§] | |
| InternVL3 8B[§] | InternVL3-8B | 512 | 0.0 | 1.0 | | 32 frames[§] | |
| InternVL2 40B[¶] | InternVL2-40B | 512 | 0.0 | 1.0 | | 8 frames[¶] | |
| InternVL2 8B[§] | InternVL2-8B | 512 | 0.0 | 1.0 | | 32 frames[§] | |

## A.2. Implementation Details for Model Inference

All MFMs are evaluated in a zero-shot setting across all tasks, consistent with prior work (Wang et al., 2024b; Shangguan et al., 2025; Zhao et al., 2025; Yue et al., 2025; Yang et al., 2025b). Whenever possible, we use the official code provided by each model for video preprocessing. Proprietary models and Qwen-series models are evaluated via their official API services[1]. In the evaluation prompt (§ B.2), models are instructed to return responses in a JSON-like format that includes both the selected multiple-choice option and the corresponding reasoning trace, enabling structured analysis and automatic parsing. When standard regular-expression–based parsing fails, we employ GPT-4o-mini to extract the multiple-choice answers; the extraction prompt is provided in § B.3.

For thinking-enabled models, the thinking_level parameter for Gemini 3 Flash, Gemini 2.5 Pro, and Gemini 2.5 Flash is set to the default dynamic mode[2], allowing the model to adapt its reasoning budget based on task complexity. For Gemini 3 Pro, thinking_level is set to low, as the dynamic setting consistently produces reasoning traces that exceed the output context window, leading to incomplete or null responses. For GPT-5.2, the thinking level is set to medium. For the Qwen3-VL series, the thinking variants are used instead of the instruct variants, with thinking_level set to medium.

---

[1] OpenAI, Gemini, and Qwen-series

[2] Gemini thinking mode

# B. Prompts

## B.1. System Prompt

---

### System Prompt

You are the person wearing the AR glasses. All videos are recorded from your first-person perspective. Treat the camera's movement as your own head and body movement. Only reason about what is visible or inferable from the egocentric video. Do not assume any external knowledge beyond what appears in the video.

---

## B.2. Evaluation Prompt

We follow the evaluation prompt used in TOMATO (Shangguan et al., 2025).

---

### Evaluation Prompt

You will be provided with a sequence of frames uniformly sampled from a video, the frames are provided in chronological order of the video. Analyze these frames and provide the answer to the question about the video content. Answer the multiple-choice question about the video content.

You must use these frames to answer the multiple-choice question; do not rely on any external knowledge or commonsense.

```
<question>
{question}
</question>

<options>
{index2ans}
</options>
```

PLEASE ANSWER THE QUESTION WITH ONLY THE OPTIONS PROVIDED. When answering, please follow the template provided:

```
"options": <your choice>
"thinking_trace": <your thinking trace>
```

---

## B.3. Answer Extraction Prompt

We follow the answer extraction prompt used in TOMATO (Shangguan et al., 2025).

---

**Answer Extraction Prompt**

```
You are given a response, a list of multiple-choice options, and an index2answer mapping. You are
    required to extract the letter option from GPT.

<response>
{response}
</response>

<all_choices>
{all_choices}
</all_choices>

<index2answer>
{index2ans}
</index2answer>

Only output the single parsed letter from the response. No other text is needed.

If you think no options can match the index2answer dictionary, randomly select one letter.

Your extracted letter is:
```

---

# C. Baselines

We summarize the baseline configurations in Section C. Additional details for the Blind LLM baseline are provided in § C.1, the Socratic model in § C.2, and the human evaluation in § C.3.

## C.1. Blind LLM

Blind LLM refers to a language-only model that does not receive any visual input, such as images or videos, during the inference stage. Instead, the model is provided solely with the textual component of the task, including the question, instructions, and answer options. This setting isolates the contribution of linguistic priors and textual reasoning, serving as a diagnostic baseline for identifying language-only shortcuts and estimating models' performance without perceptual information. By comparison with vision-enabled settings, this baseline quantifies the extent to which visual information contributes to task performance. We follow the same blind LLM prompt used in HourVideo (Chandrasegaran et al., 2024).

---

**Blind LLM Prompt**

```
You are tasked with assisting in answering a few difficult questions about short egocentric videos.
    The goal is to establish a baseline for how many multiple-choice questions (MCQs) can be
    accurately answered without watching the videos. This may involve identifying poorly crafted
    distractor options or leveraging general knowledge and logical reasoning when the questions
    themselves are straightforward.

You are STRICTLY expected to choose the correct MCQ answer based on your best judgment and provide a
    one-line reason for your selection.

<question>
{question}
</question>

<options>
{index2ans}
</options>

DO NOT GENERATE ANSWER SUCH AS 'NOT POSSIBLE TO DETERMINE.'
```

---

## C.2. Socratic Model

The Socratic model (Zeng et al., 2023) is a framework in which visual perception and textual reasoning are decoupled through a two-stage pipeline. In the first stage, for a video question-answering task, a video captioning model converts the visual input into a language-only caption. In the second stage, a language model receives the generated caption together with the question and answer options, and performs reasoning entirely in the textual domain. Under this setting, the reasoning model does not access raw visual inputs; instead, all perceptual information is compressed through the caption. This framework evaluates how informative the visual input remains when mediated solely through textual descriptions.

In our baseline evaluation, we use GPT-5.2 (Singh et al., 2026) for both video captioning and caption-based question answering. Videos are sampled at 2 fps, and the sampled frames are stitched into $4 \times 4$ image grids, with each grid containing 16 frames. For each video, multiple $4 \times 4$ grids are provided to the captioning model to generate a textual description. We follow the same caption generation and evaluation prompts used in HourVideo (Chandrasegaran et al., 2024). Caption generation prompt is provided in §C.2.1. Evaluation prompt using video caption is provided in §C.2.2. Example input for caption generation is provided in §C.2.3.

## C.2.1. CAPTION GENERATION PROMPT

> ### Caption Generation Prompt
>
> ```
> MAIN INSTRUCTIONS:
>
> Your task is to analyze video frames extracted uniformly from a short ego-centric video for a
>     detailed video understanding exercise. I will provide a sequence of images sampled at 2 frames
>     per second (2 fps) from this video. Examine the video frames closely and generate a comprehensive
>      caption by strictly following the steps below:
>
> Step 1: **Scene Context**
> Observe the frames. What is the primary setting and activity in the video?
>
> Step 2: **Motion Description**
> Identify and describe any significant motion or actions taking place across the frames.
>
> Step 3: **Spatial Relationship Analysis**
> Examine and report the spatial relationships between key objects or entities. Describe the
>     positioning and orientation of each element relative to others.
>
> Step 4: **Detailed Object Analysis**
> List the key objects and entities. Describe visible attributes such as color, shape, texture, and
>     other notable features with precision (e.g., materials, signage text, tool types, object parts).
>
> Step 5: **Temporal Relationship Context**
> Describe any observable temporal progression or changes across the sequence (e.g., before/after
>     changes, object movement, state changes, action sequences). If no meaningful change is visible,
>     state that explicitly.
>
> Step 6: **Additional Detail-Oriented Observations**
> Add any other concrete, detail-oriented observations that could help answer fine-grained questions
>     later (e.g., small objects, relative distances, occlusions, left/right placement, openings/
>     closures, item locations), but do not speculate beyond what is visible.
>
> Step 7: **Summary**
> Provide a concise yet comprehensive summary capturing the key elements from Steps 1-6.
>
> GUIDELINES:
>
> 1. Return your results in a paragraph format with the following fields:
>    - Scene Context
>    - Motion Description
>    - Spatial Relationship Analysis
>    - Detailed Object Analysis
>    - Temporal Relationship Context
>    - Additional Details
>    - Summary
>
> 2. The total length of the output must not exceed 200 words.
> ```

## C.2.2. SOCRATIC MODEL EVALUATION PROMPT

---

**Socratic Model Evaluation Prompt**

You will be provided with a textual caption that describes the content of a video. The caption is
    derived from the video and reflects its observable visual and spatial information. Analyze the
    caption and answer the multiple-choice question about the video content.

You must use only the information contained in the provided caption to answer the question; do not
    rely on any external knowledge, assumptions, or commonsense beyond what is explicitly stated in
    the caption.

<video_caption>
{video_caption}
</video_caption>

<question>
{question}
</question>

<options>
{index2ans}
</options>

PLEASE ANSWER THE QUESTION USING ONLY THE OPTIONS PROVIDED. When answering, strictly follow the
    template below:

"options": <your choice>
"thinking_trace": <your reasoning based solely on the caption>

---

## C.2.3. EXAMPLE INPUTS FOR THE SOCRATIC MODEL BASELINE

We provide an example input for caption generation in Figure 8.

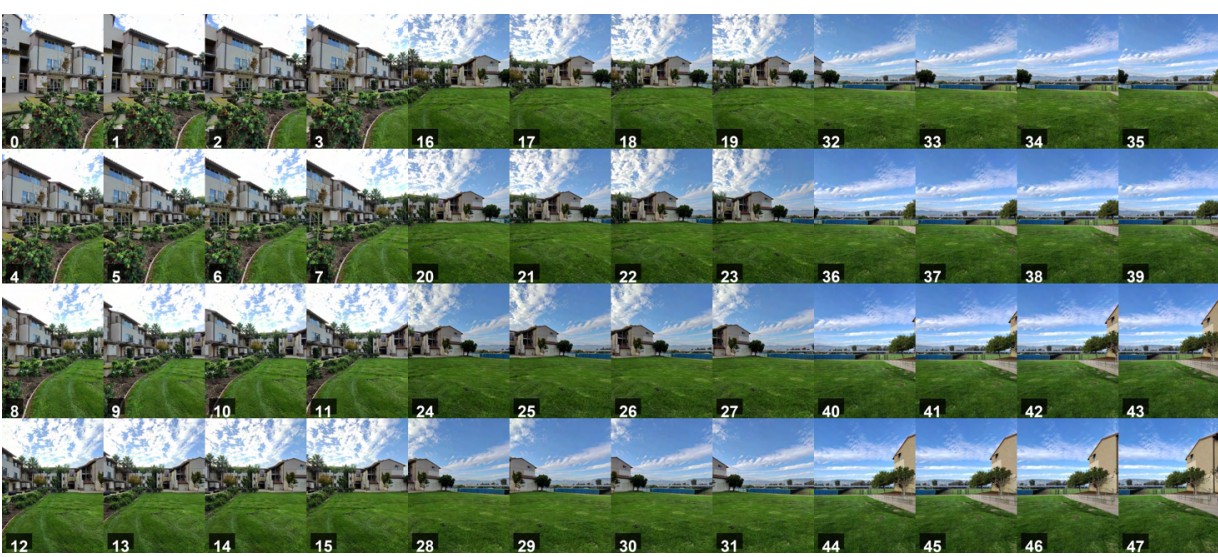

*Figure 8.* **Example Input to the Socratic Baseline.** Each $4 \times 4$ grid contains 16 frames sampled at 2 fps, with frame indices shown in the bottom-left corner of each frame.

## C.3. Human Evaluation

To establish a human performance upper bound on the benchmark, two graduate students were recruited to complete the full set of evaluation tasks. Each annotator answers all questions independently, with unlimited time and full access to the corresponding videos. Annotators are instructed to rely solely on the visual information provided, without external tools or discussion. A breakdown of individual annotator performance is reported in Table 5. Human evaluation interface is shown in Figure 9.

*Table 5.* **Human Baseline Performance on SAW-BENCH.**

| Model | All | Self-Localization | Relative Direction | Route Shape | Reverse Route Plan | Spatial Memory | Spatial Affordance |
|---|---|---|---|---|---|---|---|
| Annotator 1 | 91.07 | 98.50 | 86.69 | 98.53 | 91.70 | 94.00 | 76.54 |
| Annotator 2 | 92.03 | 89.50 | 92.09 | 96.70 | 94.32 | 83.00 | 81.48 |
| **Average** | **91.55** | **94.00** | **89.39** | **97.62** | **93.01** | **88.50** | **79.01** |

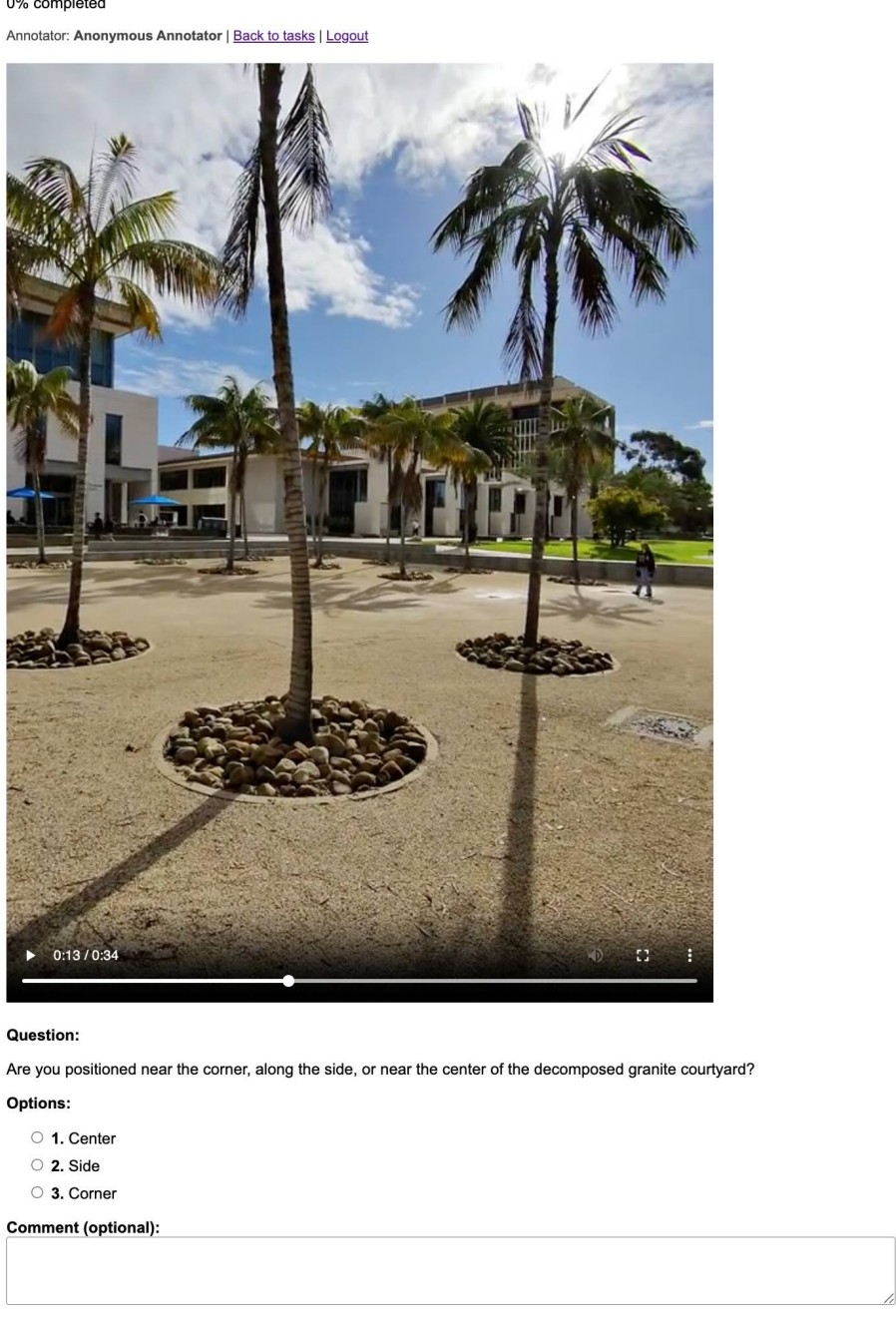

**Question:**

Are you positioned near the corner, along the side, or near the center of the decomposed granite courtyard?

**Options:**

○ **1.** Center
○ **2.** Side
○ **3.** Corner

**Comment (optional):**

Submit & Next

*Figure 9.* **Human Evaluation Interface.** Each annotator answers all questions independently, with unlimited time and full access to the corresponding videos. Annotators are instructed to rely solely on the visual information provided, without external tools or discussion.

# D. Video Filming Protocol and Meta Information Annotation

## D.1. Video Filming Protocol

We define a structured recording protocol to ensure consistent coverage of observer-centric spatial reasoning primitives while maintaining controllable trajectory complexity. Each video is associated with a predefined movement pattern, a set of spatial queries, and deterministic ground-truth answers derived from the recording plan.

We divide the recording protocol into four trajectory categories: (1) **In-place orientation** (§D.1.1), in which the camera wearer remains at a fixed spatial location and only rotates their viewpoint; (2) **Manhattan-style piecewise linear trajectories** (§D.1.2), in which the camera wearer follows a predefined path with two turns; (3) **Simple geometric trajectories** (§D.1.3), in which the camera wearer moves along canonical geometric paths; and (4) **Extra video collections** (§D.1.4), which include additional recordings designed to support the Spatial Memory and Spatial Affordance tasks.

### D.1.1. IN-PLACE ORIENTATION

*Table 6.* **Video Filming Protocol for In-Place Orientation.** Camera wearers remain at a fixed spatial location (*i.e.,* corner, side, or center of a scene) and only changes their viewing orientation, without introducing translational motion. Whenever possible, we include instances from each spatial location to ensure coverage across different positional contexts. For each video, we asked one Self-Localization and one Route Shape question.

| Spatial Location | Video ID | Q1 | A1 | Q2 | A2 |
|---|---|---|---|---|---|
| Corner 1 | 0 | Self-Localization: Are you positioned near the corner, along the side, or near the center of the ⟨Scene Name⟩? | Corner | Route Shape: What's the shape of my moving trajectory? | In-place orientation |
| Corner 2 | 1 | Self-Localization: Are you positioned near the corner, along the side, or near the center of the ⟨Scene Name⟩? | Corner | Route Shape: What's the shape of my moving trajectory? | In-place orientation |
| Corner 3 | 2 | Self-Localization: Are you positioned near the corner, along the side, or near the center of the ⟨Scene Name⟩? | Corner | Route Shape: What's the shape of my moving trajectory? | In-place orientation |
| Corner 4 | 3 | Self-Localization: Are you positioned near the corner, along the side, or near the center of the ⟨Scene Name⟩? | Corner | Route Shape: What's the shape of my moving trajectory? | In-place orientation |
| Side 1 | 4 | Self-Localization: Are you positioned near the corner, along the side, or near the center of the ⟨Scene Name⟩? | Side | Route Shape: What's the shape of my moving trajectory? | In-place orientation |
| Side 2 | 5 | Self-Localization: Are you positioned near the corner, along the side, or near the center of the ⟨Scene Name⟩? | Side | Route Shape: What's the shape of my moving trajectory? | In-place orientation |
| Side 3 | 6 | Self-Localization: Are you positioned near the corner, along the side, or near the center of the ⟨Scene Name⟩? | Side | Route Shape: What's the shape of my moving trajectory? | In-place orientation |
| Side 4 | 7 | Self-Localization: Are you positioned near the corner, along the side, or near the center of the ⟨Scene Name⟩? | Side | Route Shape: What's the shape of my moving trajectory? | In-place orientation |
| Center | 8 | Self-Localization: Are you positioned near the corner, along the side, or near the center of the ⟨Scene Name⟩? | Center | Route Shape: What's the shape of my moving trajectory? | In-place orientation |

*Table 7.* **Video Filming Protocol for Manhattan-Style Piecewise Linear.** Observers' moving trajectories are visualized using egocentric action arrows. For each video, we asked two **Relative Direction** questions, one **Reverse Route Plan** question, and one **Route Shape** question.

| Trajectory | Video ID | Q1 | A1 | Q2 | A2 | Q3 | A3 |
|---|---|---|---|---|---|---|---|
| ↑ ← ↑ | 18 | **Relative Direction**: From my viewpoint at the ⟨beginning / end⟩ of this video, where am I located at the ⟨end / beginning⟩ of the video? | ⟨front left / back right⟩ | **Reverse Route Plan**: From my viewpoint at the end of the video, how can I go back to my starting point? | Turn around, then go straight. Turn **left**, go straight, then turn **right** and continue straight. | **Route Shape**: What's the shape of my moving trajectory? | Manhattan-style piecewise linear |
| ← ↓ ↑ | 19 | **Relative Direction**: From my viewpoint at the ⟨beginning / end⟩ of this video, where am I located at the ⟨end / beginning⟩ of the video? | ⟨ left / left ⟩ | **Reverse Route Plan**: From my viewpoint at the end of the video, how can I go back to my starting point? | Turn around, go straight. Turn **right**, go straight, then turn **right** and continue go straight. | **Route Shape**: What's the shape of my moving trajectory? | Manhattan-style piecewise linear |
| → ↑ | 20 | **Relative Direction**: From my viewpoint at the ⟨beginning / end⟩ of this video, where am I located at the ⟨end / beginning⟩ of the video? | ⟨ front right / back left⟩ | **Reverse Route Plan**: From my viewpoint at the end of the video, how can I go back to my starting point? | Turn around, go straight. Turn **right**, go straight, then turn **left** and continue go straight. | **Route Shape**: What's the shape of my moving trajectory? | Manhattan-style piecewise linear |
| → ↑ ↓ | 21 | **Relative Direction**: From my viewpoint at the ⟨beginning / end⟩ of this video, where am I located at the ⟨end / beginning⟩ of the video? | ⟨ right / right ⟩ | **Reverse Route Plan**: From my viewpoint at the end of the video, how can I go back to my starting point? | Turn around, go straight. Turn **left**, go straight, then turn **left** and continue go straight. | **Route Shape**: What's the shape of my moving trajectory? | Manhattan-style piecewise linear |
| ← ↑ ← | 22 | **Relative Direction**: From my viewpoint at the ⟨beginning / end⟩ of this video, where am I located at the ⟨end / beginning⟩ of the video? | ⟨front left / back left⟩ | **Reverse Route Plan**: From my viewpoint at the end of the video, how can I go back to my starting point? | Turn around, go straight. Turn **right**, go straight, then turn **left** and continue go straight. | **Route Shape**: What's the shape of my moving trajectory? | Manhattan-style piecewise linear |
| → ↑ ← | 23 | **Relative Direction**: From my viewpoint at the ⟨beginning / end⟩ of this video, where am I located at the ⟨end / beginning⟩ of the video? | ⟨front / right⟩ | **Reverse Route Plan**: From my viewpoint at the end of the video, how can I go back to my starting point? | Turn around, go straight. Turn **left**, go straight, then turn **left** and continue go straight. | **Route Shape**: What's the shape of my moving trajectory? | Manhattan-style piecewise linear |
| ← ↓ ← | 24 | **Relative Direction**: From my viewpoint at the ⟨beginning / end⟩ of this video, where am I located at the ⟨end / beginning⟩ of the video? | ⟨ back left / back right⟩ | **Reverse Route Plan**: From my viewpoint at the end of the video, how can I go back to my starting point? | Turn around, go straight. Turn **left**, go straight, then turn **right** and continue go straight. | **Route Shape**: What's the shape of my moving trajectory? | Manhattan-style piecewise linear |
| ← ↓ → | 25 | **Relative Direction**: From my viewpoint at the ⟨beginning / end⟩ of this video, where am I located at the ⟨end / beginning⟩ of the video? | ⟨back / left⟩ | **Reverse Route Plan**: From my viewpoint at the end of the video, how can I go back to my starting point? | Turn around, go straight. Turn **right**, go straight, then turn **right** and continue go straight. | **Route Shape**: What's the shape of my moving trajectory? | Manhattan-style piecewise linear |
| ← ↑ → | 26 | **Relative Direction**: From my viewpoint at the ⟨beginning / end⟩ of this video, where am I located at the ⟨end / beginning⟩ of the video? | ⟨back / left⟩ | **Reverse Route Plan**: From my viewpoint at the end of the video, how can I go back to my starting point? | Turn around, go straight. Turn **right**, go straight, then turn **right** and continue go straight. | **Route Shape**: What's the shape of my moving trajectory? | Manhattan-style piecewise linear |
| → ↑ → | 27 | **Relative Direction**: From my viewpoint at the ⟨beginning / end⟩ of this video, where am I located at the ⟨end / beginning⟩ of the video? | ⟨front right / back right⟩ | **Reverse Route Plan**: From my viewpoint at the end of the video, how can I go back to my starting point? | Turn around, go straight. Turn **left**, go straight, then turn **right** and continue go straight. | **Route Shape**: What's the shape of my moving trajectory? | Manhattan-style piecewise linear |
| → ↓ ← | 28 | **Relative Direction**: From my viewpoint at the ⟨beginning / end⟩ of this video, where am I located at the ⟨end / beginning⟩ of the video? | ⟨back / left⟩ | **Reverse Route Plan**: From my viewpoint at the end of the video, how can I go back to my starting point? | Turn around, go straight. Turn **left**, go straight, then turn **left** and continue go straight. | **Route Shape**: What's the shape of my moving trajectory? | Manhattan-style piecewise linear |
| → ↓ → | 29 | **Relative Direction**: From my viewpoint at the ⟨beginning / end⟩ of this video, where am I located at the ⟨end / beginning⟩ of the video? | ⟨back right / back left⟩ | **Reverse Route Plan**: From my viewpoint at the end of the video, how can I go back to my starting point? | Turn around, go straight. Turn **right**, go straight, then turn **left** and continue go straight. | **Route Shape**: What's the shape of my moving trajectory? | Manhattan-style piecewise linear |
| ↑ ↓ ← | 30 | **Relative Direction**: From my viewpoint at the ⟨beginning / end⟩ of this video, where am I located at the ⟨end / beginning⟩ of the video? | ⟨ left / right ⟩ | **Reverse Route Plan**: From my viewpoint at the end of the video, how can I go back to my starting point? | Turn around, go straight. Turn **left**, go straight, then turn **left** and continue go straight. | **Route Shape**: What's the shape of my moving trajectory? | Manhattan-style piecewise linear |
| ← ↓ | 31 | **Relative Direction**: From my viewpoint at the ⟨beginning / end⟩ of this video, where am I located at the ⟨end / beginning⟩ of the video? | ⟨ back left / back left⟩ | **Reverse Route Plan**: From my viewpoint at the end of the video, how can I go back to my starting point? | Turn around, go straight. Turn **right**, go straight, then turn **left** and continue go straight. | **Route Shape**: What's the shape of my moving trajectory? | Manhattan-style piecewise linear |
| ↓ → ↑ | 32 | **Relative Direction**: From my viewpoint at the ⟨beginning / end⟩ of this video, where am I located at the ⟨end / beginning⟩ of the video? | ⟨ right / left ⟩ | **Reverse Route Plan**: From my viewpoint at the end of the video, how can I go back to my starting point? | Turn around, go straight. Turn **right**, go straight, then turn **right** and continue go straight. | **Route Shape**: What's the shape of my moving trajectory? | Manhattan-style piecewise linear |
| ↓ → ↓ | 33 | **Relative Direction**: From my viewpoint at the ⟨beginning / end⟩ of this video, where am I located at the ⟨end / beginning⟩ of the video? | ⟨back right / back right⟩ | **Reverse Route Plan**: From my viewpoint at the end of the video, how can I go back to my starting point? | Turn around, go straight. Turn **left**, go straight, then turn **right** and continue go straight. | **Route Shape**: What's the shape of my moving trajectory? | Manhattan-style piecewise linear |

## D.1.3. SIMPLE SHAPE TRAJECTORIES

*Table 8.* **Video Filming Protocol for Other Simple Trajectories.** For each video, we asked two Relative Direction questions and one Route Shape question.

| Trajectory | Video ID | Q1 | A1 | Q2 | A2 |
|---|---|---|---|---|---|
| Straight | 34 | Relative Direction: From my viewpoint at the ⟨beginning / end⟩ of this video, where am I located at the ⟨end / beginning⟩ of the video? | ⟨left / right⟩ | Route Shape: What's the shape of my moving trajectory? | Straight |
| Straight w/ turning head frequently | 35 | Relative Direction: From my viewpoint at the ⟨beginning / end⟩ of this video, where am I located at the ⟨end / beginning⟩ of the video? | ⟨front / back⟩ | Route Shape: What's the shape of my moving trajectory? | Straight |
| → ↑ (L-Shape) | 36 | Relative Direction: From my viewpoint at the ⟨beginning / end⟩ of this video, where am I located at the ⟨end / beginning⟩ of the video? | ⟨front right / back right⟩ | Route Shape: What's the shape of my moving trajectory? | L-Shape |
| ← ↑ (L-Shape) | 37 | Relative Direction: From my viewpoint at the ⟨beginning / end⟩ of this video, where am I located at the ⟨end / beginning⟩ of the video? | ⟨front left / back left⟩ | Route Shape: What's the shape of my moving trajectory? | L-Shape |
| ⊂ (U-Shape) | 38 | Relative Direction: From my viewpoint at the ⟨beginning / end⟩ of this video, where am I located at the ⟨end / beginning⟩ of the video? | ⟨front / right⟩ | Route Shape: What's the shape of my moving trajectory? | U-Shape |
| ⊃ (U-Shape) | 39 | Relative Direction: From my viewpoint at the ⟨beginning / end⟩ of this video, where am I located at the ⟨end / beginning⟩ of the video? | ⟨back / right⟩ | Route Shape: What's the shape of my moving trajectory? | U-Shape |
| ∪ (U-Shape) | 40 | Relative Direction: From my viewpoint at the ⟨beginning / end⟩ of this video, where am I located at the ⟨end / beginning⟩ of the video? | ⟨right / left⟩ | Route Shape: What's the shape of my moving trajectory? | U-Shape |
| ∩ (U-Shape) | 41 | Relative Direction: From my viewpoint at the ⟨beginning / end⟩ of this video, where am I located at the ⟨end / beginning⟩ of the video? | ⟨right / right⟩ | Route Shape: What's the shape of my moving trajectory? | U-Shape |
| Circle (Clockwise) | 42 | Relative Direction: From my viewpoint at the ⟨beginning / end⟩ of this video, where am I located at the ⟨end / beginning⟩ of the video? | ⟨same location / same location⟩ | Route Shape: What's the shape of my moving trajectory? | Circular |
| Circle (Counter-clockwise) | 43 | Relative Direction: From my viewpoint at the ⟨beginning / end⟩ of this video, where am I located at the ⟨end / beginning⟩ of the video? | ⟨same location / same location⟩ | Route Shape: What's the shape of my moving trajectory? | Circular |
| Zigzag | 44 | Relative Direction: From my viewpoint at the ⟨beginning / end⟩ of this video, where am I located at the ⟨end / beginning⟩ of the video? | ⟨front / back⟩ | Route Shape: What's the shape of my moving trajectory? | Zigzag |
| Square | 45 | Relative Direction: From my viewpoint at the ⟨beginning / end⟩ of this video, where am I located at the ⟨end / beginning⟩ of the video? | ⟨same location / same location⟩ | Route Shape: What's the shape of my moving trajectory? | Square |
| Triangle | 46 | Relative Direction: From my viewpoint at the ⟨beginning / end⟩ of this video, where am I located at the ⟨end / beginning⟩ of the video? | ⟨same location / same location⟩ | Route Shape: What's the shape of my moving trajectory? | Triangle |

## D.1.4. EXTRA VIDEO COLLECTIONS

In addition to the predefined observer trajectories described in §D.1.1, §D.1.2, and §D.1.3, we instruct video collectors to record additional videos in each scene to support the Spatial Memory and Spatial Affordance tasks. For Spatial Memory, video collectors identify objects that have been moved or displaced within the scene over the course of the trajectory. For Spatial Affordance, video collectors assess whether a queried action is physically feasible given the scene's spatial and physical constraints.

### D.2. Meta Information Annotation

Ground-truth answers are derived directly from the predefined trajectory and viewpoint sequence, independent of visual appearance. Screenshot of annotation interface is provided in Figure 9.

**Self-Localization.** Ground-truth answers are deterministically defined by the filming location of the observer within the scene (*i.e.,* corner, side, or center). Since observer positions are predefined during data collection, labels of Self-Localization do not require human annotation.

**Relative Direction.** Ground-truth answers are annotated by human annotators based on the relative spatial relationship between the observer's starting and ending viewpoints along the trajectory. Each question is independently annotated by two annotators, and inter-annotator agreement is reported to assess annotation consistency. We annotate only trajectory-level meta information; in this way, all videos that share the same trajectory share the same relative-direction labels. The inter-annotator agreement score is reported in Table 9. Any annotation disagreements are resolved through discussion at the end of the annotation process.

**Route Shape.** Ground-truth answers are deterministically specified by the predefined movement trajectory (*e.g.,* straight, L-shape, U-shape, circular). Similar to Self-Localization, these labels are directly derived from the filming protocol and do not depend on visual interpretation.

**Reverse Route Plan.** Ground-truth answers for reverse route planning are defined analogously to Relative Direction, describing the sequence of actions required to return from the final viewpoint to the starting viewpoint. These answers are annotated by human annotators to account for egocentric action interpretation, following the same annotation protocol as relative direction. Similarly, we annotate only trajectory-level meta information. Inter-annotator agreement score is reported in Table 9. Any annotation disagreements are resolved through discussion at the end of the annotation process.

**Spatial Memory.** Ground-truth answers are specified during the filming process by intentionally moving or displacing selected objects along the trajectory. Since object state changes are controlled and recorded at capture time, no additional human annotation is required.

**Spatial Affordance.** Ground-truth answers are likewise determined during filming by designing queries whose feasibility is unambiguous given the scene's physical layout and constraint. Similar to Spatial Memory, no additional human annotation is required.

*Table 9.* **Inter-Annotator Agreement Measured by Cohen's $\kappa$.**

| Task | Cohen's $\kappa$ |
|---|---|
| Relative Direction | 0.987 |
| Reverse Route Plan | 0.952 |

# E. Data Analysis

## E.1. Video Duration Distribution

We provide video duration distribution of SAW-BENCH in Figure 10.

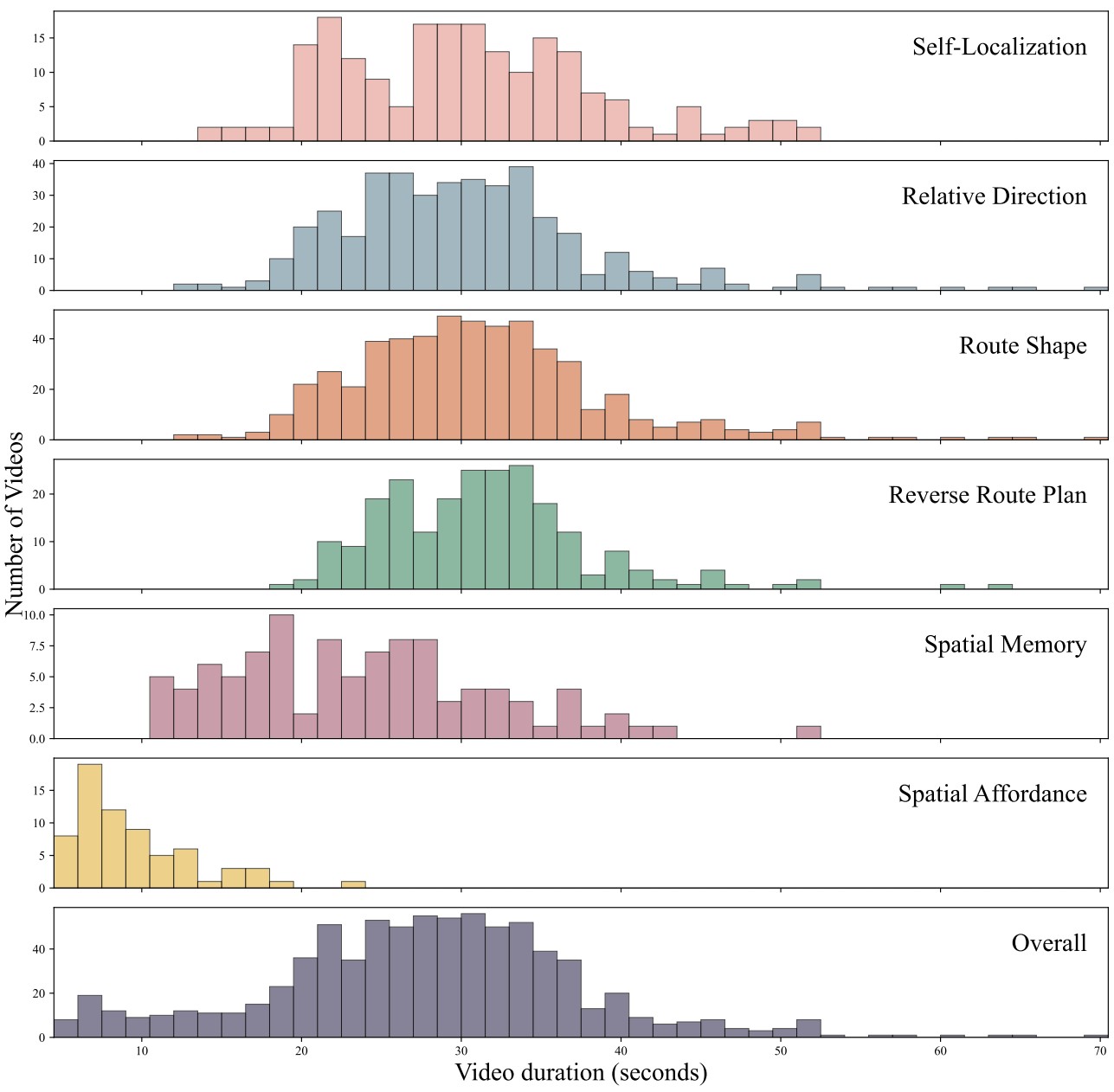

*Figure 10.* **Video duration distribution of SAW-BENCH.**

## E.2. Question Scene Distribution

We provide question scene distribution of SAW-BENCH in Figure 11.

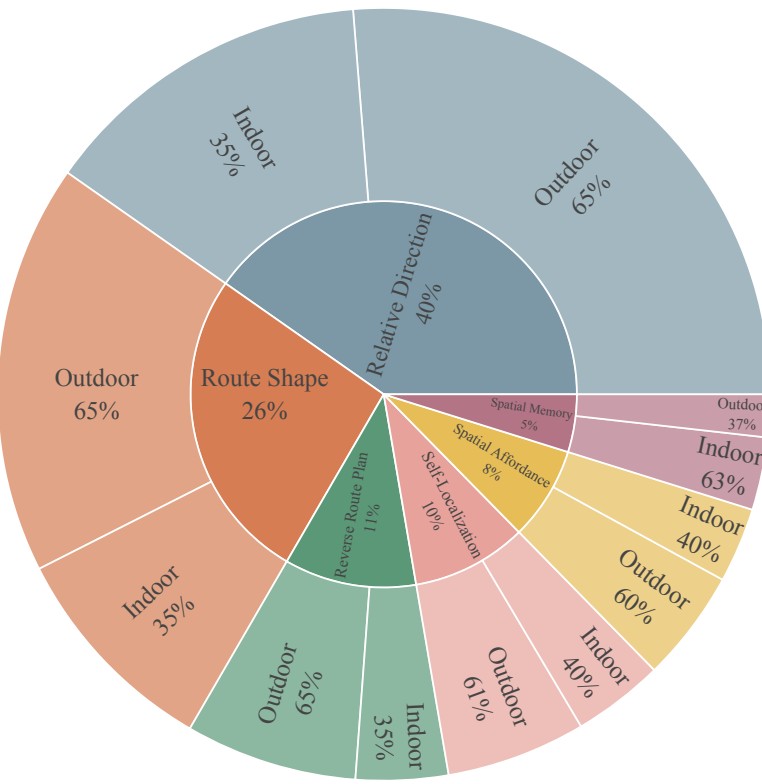

*Figure 11.* **Question scene distribution of SAW-BENCH.**

## E.3. Key Statistics

We summarize key statistics of SAW-BENCH in Table 10.

*Table 10.* **Key statistics of SAW-BENCH.**

| Statistics | Value |
|---|---:|
| Unique Videos | 786 |
|     Indoor | 317 (40.33%) |
|     Outdoor | 469 (59.67%) |
| Number of Scenes | 15 |
| Number of Videos per scene (avg) | 42 |
| Number of Videos in Unassigned Scene | 155 |
| Total Questions | 2,071 |

# F. Sensitivity Analysis

## F.1. Sensitivity to Number of Input Frames

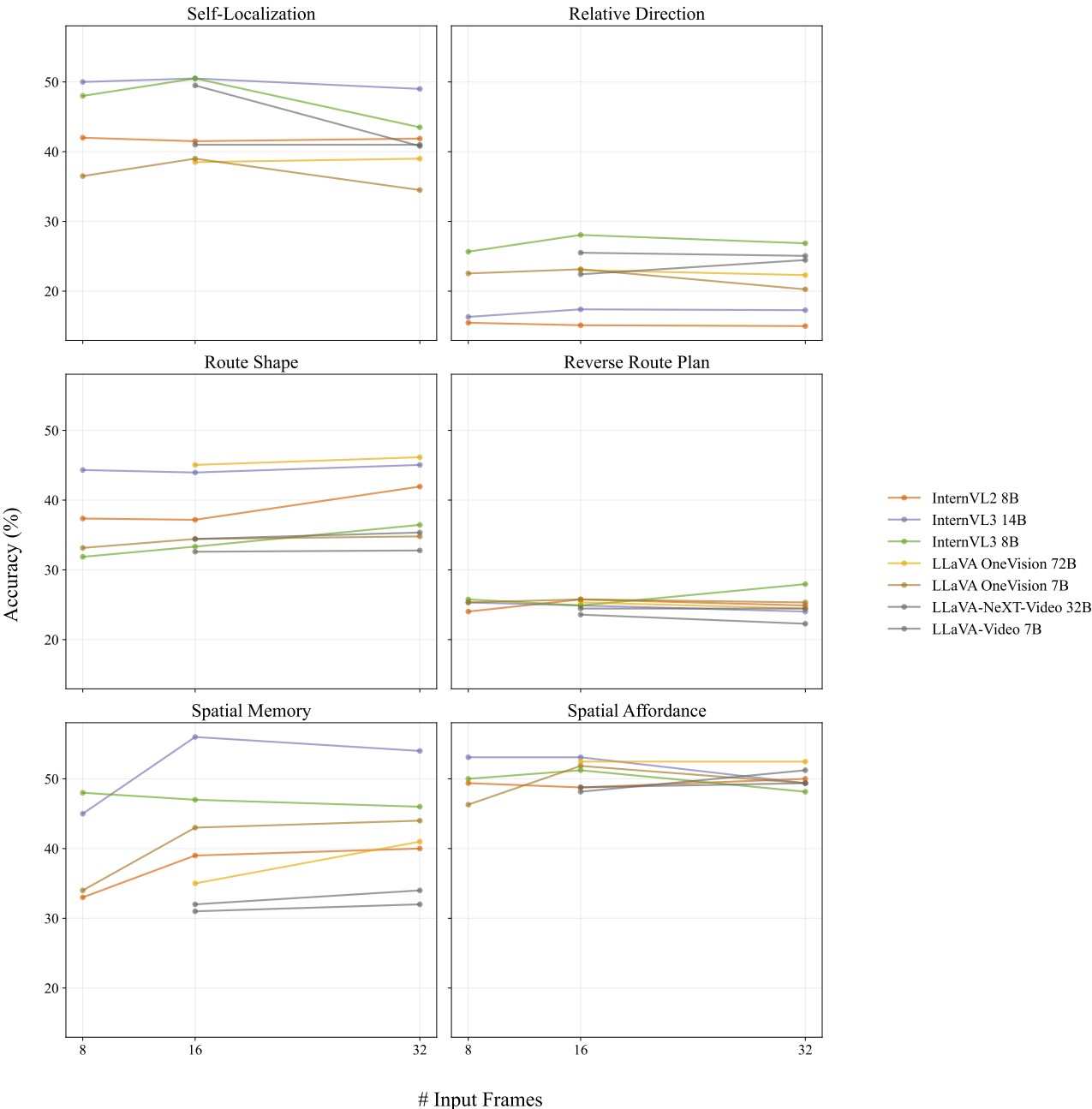

*Figure 12.* **Sensitivity to the Number of Input Frames Across Reasoning Types.** Each subplot reports accuracy as a function of the number of input frames (8, 16, 32) for a fixed reasoning type. Lines correspond to different open-source MFMs evaluated with varying temporal context lengths. Tasks that require longer-horizon spatial integration, such as **Spatial Memory** and **Route Shape**, generally benefit from increased temporal context, whereas short-horizon tasks (*e.g.,* **Self-Localization** and **Relative Direction**) exhibit earlier saturation and more limited improvements beyond 16 frames. Nevertheless, additional frames yield only marginal gains across all tasks.

          30          

## F.2. Sensitivity to Frame Sampling Rate (FPS)

*Table 11.* **Sensitivity to Frame Sampling Rate (FPS) Across Reasoning Types.** We evaluate the top-performing model, Gemini 3 Flash, on SAW-BENCH by varying the number of input frames through different temporal sampling rates (FPS $\in \{1, 2, 3, 4, 6\}$), and report performance across reasoning tasks. All other configurations are held constant, as detailed in Table 4. Overall, increasing the frame sampling rate does not substantially improve situated spatial reasoning. Notably, performance on **Spatial Memory** decreases at FPS = 3, potentially due to redundant or temporally condensed visual information, while **Reverse Route Plan** performs worst at FPS = 1, suggesting a greater reliance on temporal visual context for this reasoning type.

| Model | FPS | All | Self-Localization | Relative Direction | Route Shape | Reverse Route Plan | Spatial Memory | Spatial Affordance |
|---|---|---|---|---|---|---|---|---|
| Gemini 3 Flash | 1 | 53.55 | 45.00 | 39.93 | 69.05 | 55.90 | 64.00 | 72.22 |
| Gemini 3 Flash | 2 | 53.89 | 48.50 | 41.13 | 64.84 | 61.57 | 66.00 | 70.99 |
| Gemini 3 Flash | 3 | 51.57 | 42.50 | 40.29 | 60.99 | 63.76 | 53.00 | 70.99 |
| Gemini 3 Flash | 4 | 52.25 | 47.50 | 39.93 | 63.55 | 65.50 | 55.00 | 62.96 |
| Gemini 3 Flash | 6 | 49.25 | 40.50 | 39.33 | 56.96 | 59.83 | 54.00 | 67.28 |

# G. Common Failure Cases

## G.1. Self-Localization

### G.1.1. EXAMPLE 36

**Gemini 3 Flash**
"options": A
"thinking_trace": The video begins with a view of the lecture hall from the back, positioned towards the right side. As the camera moves, it approaches the right-side wall, where an exit door is located. The camera then pans across the room from this side position, showing the rows of seats and the front of the hall from a side perspective. Throughout the video, the camera remains along the side or near the back corner, never moving toward the center of the hall. Therefore, the position is along the side.

**Gemini 3 Pro**
Based on the video frames, here is the analysis of my position:
1. **Initial View (00:00.000 - 00:06.500):** The video starts with a view of the lecture hall seating. The rows of chairs are to my left, and a wall is immediately to my right. I am walking along an aisle that runs parallel to the wall on the right side of the room. This clearly places me on the "side" of the seating area, not in the center or a corner.
2. **Walking Path (00:06.500 - 00:12.000):** I continue walking straight along this side aisle, passing an exit door on my right. The perspective of the rows of seats confirms I am at the edge of the seating arrangement.
3. **Turning and Looking Back (00:12.000 - 00:35.500):** I approach a set of glass doors, look through them into a lobby/atrium area, and then turn around to face the lecture hall again. When I turn back (around 00:27.000 onwards), I am looking down the same side aisle I just walked up. The screen is far to the left, and the wall is now on my left. This confirms my location is along the side wall of the room.
Therefore, throughout the video, I am positioned along the side aisle of the lecture hall.
"options": A
"thinking_trace": The video begins with the camera moving along an aisle with a wall immediately to the right and rows of seats to the left. This indicates a position on the side of the room. The camera continues down this path, passing an exit sign on the right wall. Eventually, the camera turns around and looks back down the length of the room, still positioned next to the wall, confirming the location is along the side of the lecture hall rather than in the center or tucked into a corner.

**Gemini 2.5 Pro**
"options": A

"thinking_trace": The video begins with a view across many rows of empty seats in a lecture hall, with a large window wall on the far side. I then pan my head to the right. As I pan, it becomes clear that I am standing in an aisle. To my left are the rows of seats, and to my right is a solid wall. I continue panning right along this wall until I reach a set of double doors with an 'EXIT' sign above them. I look through the glass doors for a moment before panning back to the left, retracing my view along the side wall and back towards the front of the lecture hall. Since I am in an aisle with seats on one side and a wall with an exit door on the other, my position is along the side of the lecture hall.

**GPT-5.2**
"options": A
"thinking_trace": The view stays next to the right wall/aisle (windows and side wall with EXIT door visible), looking across rows toward the screen, indicating a side position rather than the center or a corner.

**Qwen3-VL 235B**
"options": A
"thinking_trace": The camera movement along the side of the lecture hall, passing by rows of seats and windows, indicates a side position.

**Qwen3-VL 32B**
"options": A
"thinking_trace": The camera moves along the side of the lecture hall, adjacent to the dark panel wall with the exit sign and doors, indicating a position along the side.

**Qwen3-VL 30B**
"options": A
"thinking_trace": In the frames, the camera movement shows the perspective along the rows of seats in the lecture hall. The view captures the side view of the seating arrangement, indicating the position is along the side of the lecture hall.

**Qwen2.5-VL 72B**
"options": A
"thinking_trace": The video shows a view of the lecture hall from a position that is aligned with the side wall, as evidenced by the proximity to the exit sign and the glass door leading to another area. The perspective does not show a central view of the hall or a corner view where two walls would meet. Therefore, the position is along the side.

---

**Self-Localization 59**

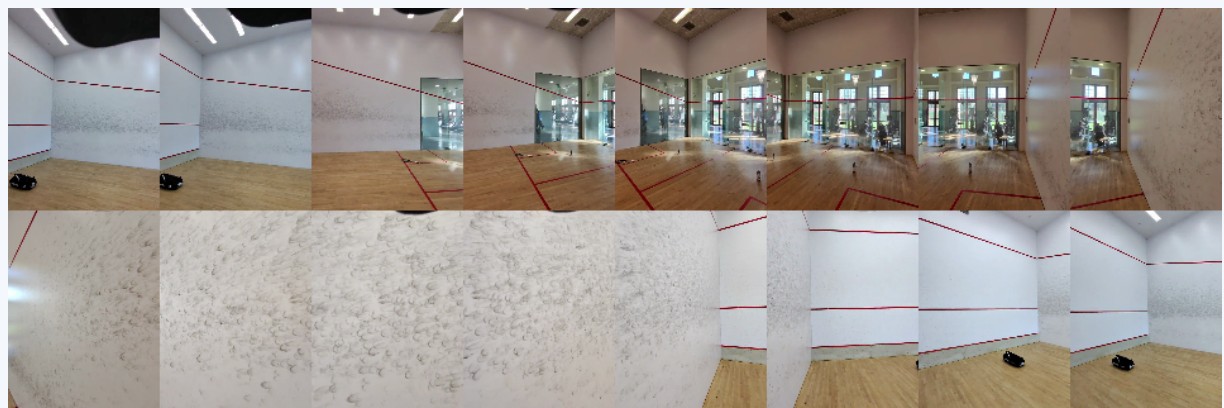

Are you positioned near the corner, along the side, or near the center of the lecture hall?

A. Center
B. Side
C. Corner

Answer: B.Side

---

**Gemini 3 Flash**
"options": C
"thinking_trace": The video begins with a view of a squash court. I can see two white walls meeting at a corner, and a wooden floor with red lines. A black bag is visible on the floor near the corner. As the camera pans, it reveals more of the court, including glass walls and people outside, but the primary focus remains on the corner of the court. The red lines on the floor, which define the service box and other court areas, are clearly visible, indicating that I am positioned within the playing area. My perspective is consistently showing the intersection of two walls and the floor, strongly suggesting I am near a corner of the court, rather than the center or just along a side.

---

**Gemini 3 Pro**
Based on the video frames, here is the analysis of my position: 1. **Initial View (00:00.000 - 00:03.000):** The video starts with a view of the front wall and the left side wall meeting at a corner. A black bag is on the floor near this corner. The perspective suggests I am standing somewhat back from the front wall, looking towards the front-left corner.
2. **Panning Right (00:03.000 - 00:13.000):** As I turn my head to the right, I see the glass back wall of the court. Through the glass, I can see a gym area with exercise equipment. The camera pans across the back wall.
3. **Focus on the Wall (00:13.000 - 00:27.000):** The camera then moves very close to the right side wall (or possibly the front wall again, but the texture suggests a side wall heavily marked with ball impacts). The view is extremely close-up, showing the scuff marks in detail. This indicates I have moved right up to a wall.
4. **Final Pan (00:27.000 - 00:32.000):** The camera pulls back and pans left again, revealing the front wall and the front-left corner where the bag is located. The angle of view clearly shows two walls meeting (front and left side).
**Conclusion:** throughout the video, the camera's perspective is anchored near the intersection of two walls. Specifically, the view starts looking into a corner, pans around, and ends looking back into that same corner (front-left). The proximity to the walls and the angle of view into the intersection of the front and side walls strongly indicate a position near the corner of the court.
"options": C
"thinking_trace": The video begins with a clear view of the intersection between the front wall and the left side wall, where a black bag is placed on the floor. This immediately establishes a perspective directed towards a corner. As the video progresses, the camera pans right to show the glass back wall and then moves in very close to a wall covered in ball marks, before panning back to the original corner view. The field of view captures the convergence of two perpendicular walls for a significant portion of the clip, and the camera movement pivots around this location. Being able to see the two walls meeting at an angle directly in front or slightly to the side is characteristic of being positioned in or near a corner. Therefore, the position is best described as near the corner.

**Gemini 2.5 Pro**
"options": A
"thinking_trace": The video begins with me looking at the front-left corner of the squash court. I then pan my head to the right, sweeping my view across the front wall, past the glass door and the back glass wall, along the right-side wall, and then back towards the front-left corner. This wide, panoramic sweep, which captures a view of all four walls from a relatively stable standing position, indicates that I am positioned somewhere in the middle of the court. If I were in a corner or along a side wall, the perspective and the way the walls appear during the pan would be significantly different and more skewed.

**GPT-5.2**
"options": C
"thinking_trace": The view repeatedly shows the junction of two walls and the floor at a corner, and the camera moves very close alongside the wall, indicating a position near the court's corner rather than centered or mid-side.

**Qwen3-VL 235B**
"options": C
"thinking_trace": The video frames show the camera moving towards and focusing on the corner where two walls of the squash court meet (evident in frames 27–29 and 30–32,which highlight the intersection of walls and the bag positioned in the corner). This indicates the person is near the corner of the court.

**Qwen3-VL 32B**
"options": C
"thinking_trace": The video frames show a consistent view of the squash court from a corner perspective. The walls and floor markings are visible in a way that indicates the camera is positioned near the corner of the court, as opposed to the center or along the side. The angle and proximity to the walls suggest a corner position.

**Qwen3-VL 30B**
"options": C
"thinking_trace": The video shows a squash court with red lines on the walls. In the initial frames, a bag is visible in the corner where two walls meet. As the camera moves, the corner area (where two walls intersect) is observed, indicating the person is near the corner of the squash court.

**Qwen2.5-VL 72B**
"options": C
"thinking_trace": The video starts with a view of the corner of the squash court, showing two walls meeting at a right angle. As the camera pans around, it captures different parts of the court but initially focuses on the corner area where a bag is placed. This indicates that the starting position is near the corner of the squash court.

## G.2. Relative Direction

### G.2.1. EXAMPLE 10

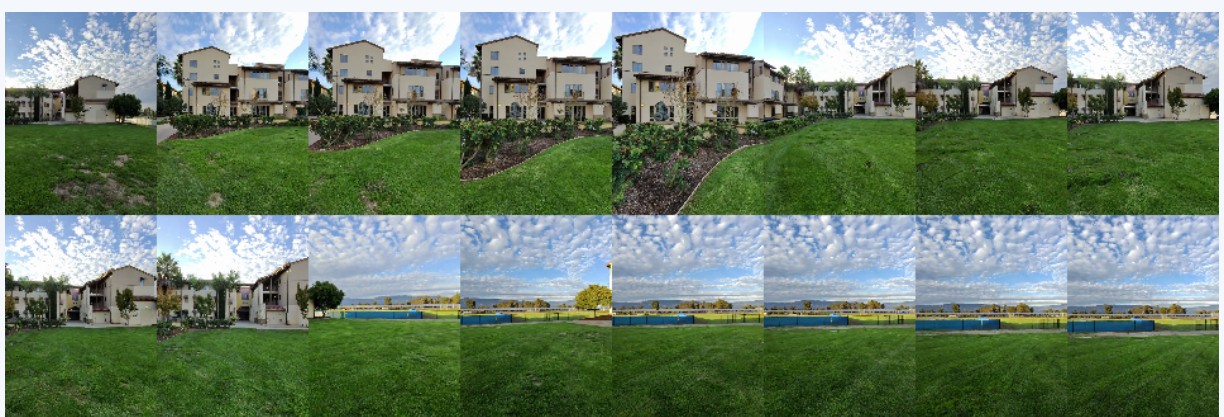

**Relative Direction 10**

From my viewing point at the beginning of this video, where am I located at the end of the video?

A. Front right
B. Front left
C. Front
D. Same location

Answer: C. Front

---

**Gemini 3 Flash**
"options": A
"thinking_trace": At the beginning of the video (00:00.000), I am standing on a lawn facing a complex of residential buildings. I begin by walking forward towards the buildings. Around 00:14.500, I turn right and walk across the lawn, parallel to the buildings. At 00:30.000, I turn right again, now facing away from the buildings and towards a sports field with a blue fence. I continue walking in this direction until the end of the video (00:46.000). Relative to my starting position and initial orientation (facing the buildings), I have moved forward and significantly to the right. Therefore, my final location is to the front right of my starting point.

---

**Gemini 3 Pro**
"options": A
"thinking_trace": At the beginning of the video, I am standing on a grassy lawn facing a large beige apartment complex. The building is directly in front of me, and I am positioned somewhat centrally relative to the courtyard area shown. As the video progresses, I begin to move forward and slightly to the right, walking across the grass. By the middle of the video (around 00:15.000), I have moved closer to the building and am now positioned more to the right side of the initial view, looking towards a breezeway or passage between the buildings. Towards the end of the video (00:30.000 onwards), I turn my head to the right and the view shifts completely away from the building to a large open field with a blue fence and mountains in the distance. However, relative to my starting position facing the building, my physical location has moved forward and to the right side of the initial scene before I turned my head. Therefore, I am located at the 'Front right' relative to my starting point.

---

**Gemini 2.5 Pro** "options": A
"thinking_trace":
1. **Initial Position (00:00):** I am standing on a large grassy area, facing a U-shaped, multi-story building. My position is relatively central on the lawn.
2. **Movement Analysis (00:00 - 00:14):** I begin by panning my head left and then walk forward and to the left, moving closer to the left wing of the building complex.
3. **Movement Analysis (00:14 - 00:30):** I then pan my head and body to the right, sweeping across the entire front of the building complex. As I pan, the perspective changes, indicating I am also moving physically to the right across the lawn.
4. **Movement Analysis (00:30 - 00:46):** My pan continues past the right side of the building, revealing a large sports field behind a fence. I stop with this field in view.

5. **Final Position (00:46):** I am now standing on the right side of the grassy area (relative to my starting point), looking at the sports field.
6. **Comparison:** I started in a central position on the lawn. I ended up on the right side of the lawn. Therefore, relative to my initial viewing point, my final location is to the front and right.

**GPT-5.2**
"options": A
"thinking_trace": I start facing the building across the lawn, then move forward toward the landscaping/sculpture and continue shifting to the right side of the courtyard, ending near the right side looking out toward the fenced sports field, so my end position is in front-right of the starting viewpoint.

**Qwen3-VL 235B**
"options": A
"thinking_trace": The video starts with a view of a building and grass. As the frames progress, the camera moves and turns, eventually showing a sports field in the final frames. From the initial viewing point, the final position corresponds to the front right direction, as the movement and rotation of the camera (from the first-person perspective) lead to a perspective that is to the right of the original viewpoint.

**Qwen3-VL 32B** "options": A
"thinking_trace": The video starts with a view of the building on the left side of the frame. By the end, the camera has moved to the right, showing the blue fence and distant mountains, which is the front right area from the initial viewing point.

**Qwen3-VL 30B**
"options": A
"thinking_trace": The video starts with a view of a building on the right side (frame 0). As the sequence progresses, the camera moves across the grassy area, showing different angles of the neighborhood. By the end of the video (frame 46), the scene shifts to a sports field with a blue fence and mountains in the background. Comparing the initial viewpoint (building on the right) to the final viewpoint (sports field), the location at the end is positioned in the front-right relative to the starting point. Thus, the answer is A.

**Qwen2.5-VL 72B**
"options": D
"thinking_trace": The video starts with a view of a building and its surroundings. As the video progresses, the camera pans around the area but does not move to a different location. The final frames show the same general area as the beginning, indicating that the viewing point has not changed significantly. Therefore, the correct answer is D: Same location.

Relative Direction 413

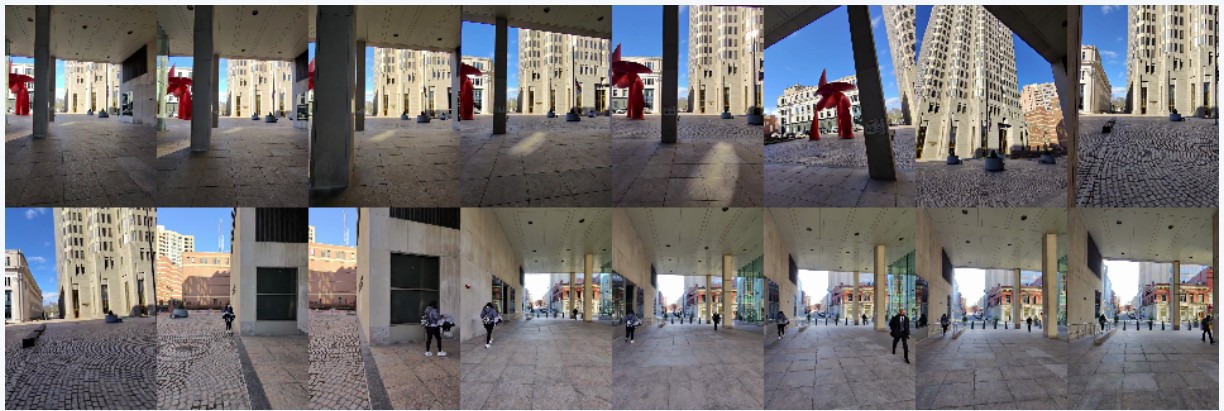

From my viewing point at the end of this video, where am I located at the beginning of the video?

A. Back right
B. Right
C. Same location
D. Front right

Answer: B. Right

---

**Gemini 3 Flash**
"options": D
"thinking_trace": At the beginning of the video (00:00.000), I am standing under a covered walkway, looking out towards a large plaza with a red sculpture and a tall skyscraper. As the video progresses, I walk out into the plaza, turn around, and then walk back under the covered walkway, but I continue walking further into it. At the end of the video (00:39.000), I am looking down the walkway towards a street. From this final viewing point, the plaza (and thus my starting location) is to my right and further ahead towards the edge of the walkway. Therefore, my starting location is to the front right relative to my final position.

---

**Gemini 3 Pro**
The camera starts under a covered walkway, looking out at a plaza with a red sculpture on the left. As the video progresses, the camera moves forward and turns to the right, walking along the covered walkway. By the end of the video, the camera is further down the walkway, looking towards a street intersection. Therefore, relative to the final position, the starting position was behind and to the right.
"options": A
"thinking_trace": The video begins with the camera positioned under a covered walkway, facing a plaza with a prominent red sculpture to the left and a tall building ahead. As the video progresses, the camera moves forward and pans right, following a person walking away. By the end of the video (around 00:39.000), the camera has advanced significantly down the walkway and is now facing a street scene with brick buildings. Looking back from this final viewpoint, the initial location (where the red sculpture was visible to the left) would be behind and to the right of the current position. Therefore, the starting location is 'Back right' relative to the ending location."

---

**Gemini 2.5 Pro**
"options": A
"thinking_trace":
1. **Analyze the initial position (00:00):** I am standing under a covered walkway, looking out into a large, open plaza. There is a tall building across the plaza and a large red sculpture to my left.
2. **Analyze the movement:** I walk forward, out from under the cover and into the plaza. I move across the plaza towards the tall building.
3. **Identify the key turn (00:21):** After crossing part of the plaza, I make a distinct turn to my right. I then begin walking along the side of the tall building, which is now on my right.
4. **Analyze the final position (00:39):** I have continued walking forward from my right turn and am now under another covered walkway along the side of the tall building. My final view is looking forward down this walkway towards a street.

5. **Determine the relative positions:** My path was an 'L' shape: I walked forward, made a 90-degree right turn, and then walked forward again.

6. **Relate the start to the end:** Imagine I am at my final position, facing the direction I am at the end of the video. To locate my starting point relative to this final position and orientation, I have to consider the path I took. The starting point is behind the corner I turned. Since I turned right to get to my final path, the initial path is to the right of my current backward direction. Therefore, from my final vantage point, the starting point is behind me and to my right.

7. **Evaluate the options:** * 'A': 'Back right' - This matches the conclusion that the starting point is behind me and to my right, relative to my final position and direction of view. * 'B': 'Right' - This would imply the starting point was directly to my side, which is incorrect as I walked forward from it. * 'C': 'Same location' - This is incorrect as I clearly moved across a plaza. * 'D': 'Front right' - This is the opposite of the correct direction.

Therefore, the correct option is 'Back right'.

---

**GPT-5.2**
"options": A
"thinking_trace": By the end I'm facing down the covered walkway toward the street; earlier I was at the plaza-side opening (with the red sculpture visible). Since I walked/turned away from that area, the beginning location is behind me and slightly to the right from the final viewpoint.

---

**Qwen3-VL 235B**
"options": A
"thinking_trace": At the end of the video (frame 40), the viewer is in a covered walkway facing a street. The beginning of the video (frame 1) shows the viewer in a different covered area (under an overhang) looking out toward a plaza with a red sculpture and a tall building. As the video progresses, the viewer moves from the initial covered area through the plaza into the final covered walkway. From the end position, the beginning location is behind the viewer, and considering the spatial orientation (the red sculpture was on the left initially, but movement direction suggests the origin is to the back right), the correct answer is A: Back right.

---

**Qwen3-VL 32B**
"options": A
"thinking_trace": The video starts with a view of a covered walkway and a large building in the background. As the camera moves, it reveals more of the surroundings, including a red sculpture and other buildings. The camera then pans to the right, showing a person walking away from the viewer. Finally, the camera returns to the initial position, indicating that the starting point is back to the right of the final viewing point.

---

**Qwen3-VL 30B**
"options": A
"thinking_trace": The video shows the camera moving from an initial position under a covered structure (frame 0) toward the end of a covered walkway (frame 39). At the end of the video, the viewing point is in the covered walkway facing the street. The beginning of the video was at the start of this journey, which corresponds to the back-right relative to the current end position due to the forward movement through the plaza and into the walkway, with the red sculpture's position on the left in early frames indicating the direction of movement.

---

**Qwen2.5-VL 72B**
"options": C
"thinking_trace": The video starts with a view from under an overhang looking out onto a plaza. As the video progresses, the camera moves forward and turns to the right, revealing more of the plaza and surrounding buildings. The final frames show a different perspective of the plaza and the street beyond. However, the starting point is still visible in the background of the final frames, indicating that the camera has not moved significantly from its original location. Therefore, the correct answer is 'C': Same location.

G.3.1. EXAMPLE 151

---

### Route Shape 151

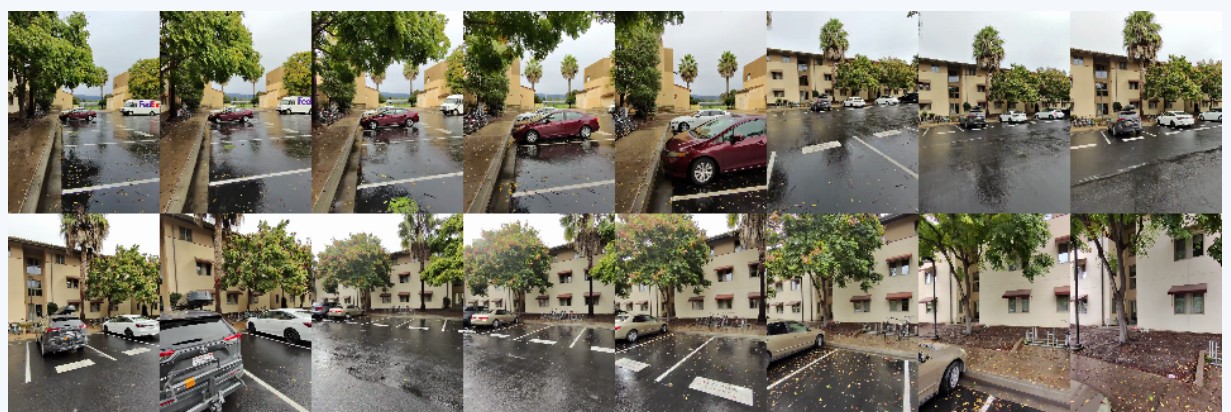

What's the shape of my moving trajectory?

A. U-Shape
B. Triangle
C. L-Shape
D. Straight line

Answer: B. Triangle

---

**Gemini 3 Flash**
"options": A
"thinking_trace": The video starts with the camera moving forward along a sidewalk next to a parking lot (00:00.000 to 00:10.500). At around 00:11.000, the camera makes a distinct right turn and continues moving forward across the parking lot (00:11.500 to 00:22.000). Then, at approximately 00:22.500, the camera makes another right turn and moves forward towards a building with bike racks (00:23.000 to 00:36.000). This sequence of two right turns connecting three straight segments forms a U-shape.

---

**Gemini 3 Pro**
The camera's movement can be broken down into two main segments based on the visual evidence:
1. **First Segment (00:00.000 - 00:22.000):** The video starts with the camera moving forward along a sidewalk next to a parking lot. We see a red car, a white car, and a FedEx truck. The camera continues straight, passing parked cars on the right (a red sedan, a white sedan, a grey SUV with a roof box, another white sedan). The movement is generally linear along the edge of the parking lot.
2. **Second Segment (00:22.500 - 00:36.000):** Around frame 00:22.500, there is a distinct change. The camera view pivots or turns significantly to the right. After this turn, the camera begins moving forward again in a new direction, perpendicular to the original path. The camera now moves towards a beige building with brown awnings, passing a gold sedan parked on the left and a bike rack on the right.
Combining these two segments:* The camera moves straight along one axis.* It then makes a roughly 90-degree turn to the right.* It continues straight along a new axis.
This pattern of movements-straight, turn 90 degrees, straight-forms an "L" shape.
"options": C
"thinking_trace": The video begins with the camera moving forward along a sidewalk parallel to a parking lot, passing several parked cars (red sedan, white sedan, grey SUV). This constitutes the first leg of the path. Around the 22-second mark, the camera stops its forward motion along this axis and turns approximately 90 degrees to the right. Following this turn, the camera proceeds forward again, now moving towards a building and passing a gold sedan and bike racks. This second leg is perpendicular to the first. The combination of a straight path, a 90-degree turn, and another straight path creates a trajectory that resembles the letter 'L'.

---

**Gemini 2.5 Pro**
"options": A

---

"thinking trace": 1. At the beginning of the video (0:00), I am standing on a sidewalk next to a parking lot. I start moving forward, parallel to a row of parked cars on my right. This is the first straight segment of my movement.
2. Around 0:10, I reach the end of the first row of cars and turn my body to the right, changing my direction of movement by approximately 90 degrees.
3. From 0:11 to 0:22, I walk across the parking lot, perpendicular to my initial path. This is the second straight segment.
4. Around 0:22, I turn my body to the left, changing my direction again by approximately 90 degrees. I am now facing parallel to my original path, but on the opposite side of the parking lot.
5. From 0:23 to the end of the video, I walk forward along this new path. This is the third straight segment.
6. The overall trajectory consists of three straight segments connected by two turns, forming a shape that resembles the letter 'U'. I walk down one side, across the middle, and then up the other side. Therefore, the shape of my moving trajectory is a U-shape.

**GPT-5.2**
"options": C
"thinking trace": The view moves forward along the left side of the parking lot toward the cars, then makes a clear right turn to face and continue along the building/parking row, forming an L-shaped trajectory.

**Qwen3-VL 235B**
"options": C
"thinking trace": The video shows movement starting straight, then a right turn (evident from the shift in direction between frames 12-13), followed by continued movement along a new straight path. This creates an L-shaped trajectory, matching option C.

**Qwen3-VL 32B**
"options": A
"thinking trace": The camera starts near the sidewalk with the red car, moves forward, then turns and the path forms a U-shape as it moves around the parking lot, indicating a U-shaped trajectory.

**Qwen3-VL 30B** "options": C
"thinking trace": To determine the shape of the moving trajectory, analyze the camera's path through the parking lot. The video shows the camera moving forward, then turning (e.g., from moving along one row of parking spaces to another perpendicular row). This pattern of moving in two perpendicular directions forms an L-Shape. Other options are inconsistent: a straight line would require no turning (not observed), U-Shape implies a complete reversal (not seen), and Triangle would require three segments (not matching the movement). Thus, the trajectory is L-Shape.

**Qwen2.5-VL 72B**
"options": C
"thinking trace": The video shows a movement starting from the left side of the parking lot, moving forward and then turning right towards the building. This trajectory forms an L-shape as it progresses from one point to another with a 90-degree turn.

Route Shape 225

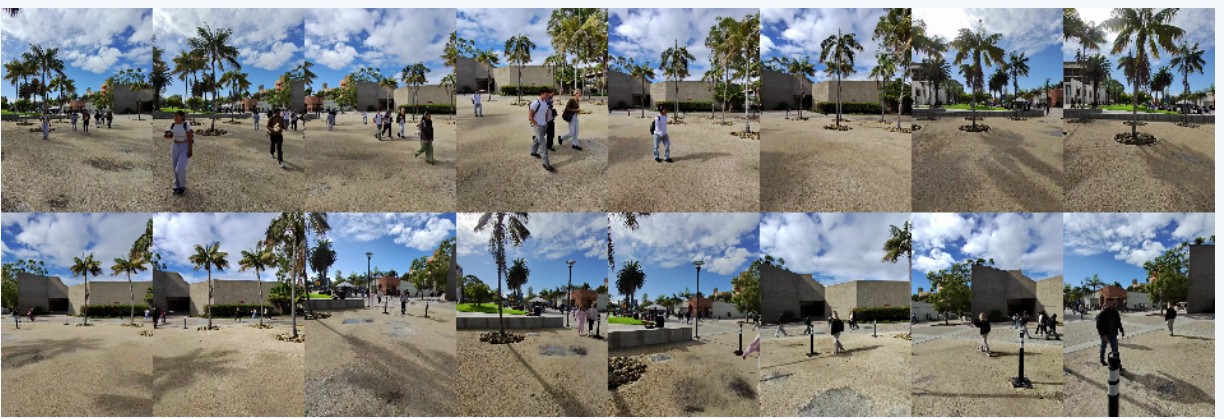

What's the shape of my moving trajectory?

A. Manhattan-style piecewise linear
B. In-place orientation
C. Straight line
D. Zigzag

Answer: D, Zigzag

**Gemini 3 Flash**
"options": A
"thinking_trace": The video shows the camera moving through an outdoor plaza. The movement consists of several distinct straight-line segments followed by sharp turns. For example, the camera moves forward, then turns left around 00:11.000, moves forward again, turns right around 00:16.000, moves forward, turns left around 00:21.000, moves forward, turns right around 00:26.000, and turns left again around 00:30.000. This pattern of moving in straight lines and making sharp, often near-right-angle turns is characteristic of a Manhattan-style piecewise linear trajectory.

**Gemini 3 Pro**
"options": A
"thinking_trace": The video starts with the camera facing a plaza with palm trees and people walking. From 00:00.000 to roughly 00:10.500, the camera seems relatively stationary or moving very slowly, mostly panning to observe the scene. Then, around 00:11.000, the camera turns left and begins to move forward along a path. This forward movement continues until about 00:20.500. At this point, the camera stops moving forward and turns right (approx 90 degrees) to face a different direction, looking towards a building labeled K̈err Hall.̈ Then, from 00:21.000 onwards, the camera moves forward again in this new direction. Finally, around 00:26.000, the camera turns left again (approx 90 degrees) and moves towards the building entrance. This pattern of moving straight, turning 90 degrees, moving straight, and turning 90 degrees again resembles a M̈anhattan-styleör grid-like movement, which is characterized by piecewise linear segments connected by sharp turns. Therefore, the trajectory is best described as Manhattan-style piecewise linear.

**Gemini 2.5 Pro**
"options": B
"thinking_trace": The video shows a first-person perspective. I start by looking forward at a group of people in a plaza with palm trees. Then, my head turns to the right, panning across the scene to focus on a large, modern, grey building. After a moment, my head pans back to the left, looking past the initial view towards a grassy area with a market tent in the distance. The camera continues to pan left and right, observing different parts of the plaza. Throughout this entire sequence, my position relative to the surrounding objects (the palm trees, the buildings, the grassy area) does not seem to change significantly. The movement is primarily rotational, as if I am standing in one spot and turning my head to look around. This is a classic example of in-place orientation change, not a trajectory involving walking or moving from one point to another. Therefore, the shape of my moving trajectory is best described as 'In-place orientation'.

**GPT-5.2**

"options": C

"thinking_trace": "Across the frames, I continuously advance forward across the plaza (objects like palm tree bases and bollards get closer), while my head pans left/right; the translation path itself appears roughly straight rather than turning in right angles or zigzagging.

**Qwen3-VL 235B**

"option": A

"thinking_trace": The video shows the perspective moving through an outdoor plaza with multiple turns at right angles, consistent with a Manhattan-style piecewise linear trajectory (moving in straight segments with 90-degree turns). The path is not a straight line (C), does not involve staying in place (B), and is not a sharp zigzag (D).

**Qwen3-VL 32B**

"options": A

"thinking_trace": The movement in the video involves changing directions multiple times, forming a path composed of several straight segments connected by turns. This matches the description of a Manhattan-style piecewise linear trajectory, where movement is along straight lines with directional changes, even if the turns aren't strictly 90-degree (as Manhattan-style can be interpreted broadly for piecewise linear paths with distinct segments). The other options don't fit: B (in-place orientation) implies no movement, C (straight line) is incorrect as direction changes, and D (zigzag) suggests more frequent back-and-forth which isn't observed.

**Qwen3-VL 30B**

"options": A

"thinking_trace": The video shows the camera moving through a space with multiple direction changes, indicating a trajectory composed of straight segments with turns. This aligns with the definition of "Manhattan-style piecewise linear" (option A), where movement follows straight paths with directional shifts rather than a single straight line (C), in-place orientation (B), or a simple zigzag (D).

**Qwen2.5-VL 72B**

"options": A

"thinking_trace": The trajectory in the video appears to be a series of straight segments that change direction at right angles, resembling a Manhattan-style grid. This is evident from the way the camera moves forward and then turns sharply at intersections or corners, maintaining a piecewise linear path throughout the sequence of frames. There are no indications of in-place orientation, a continuous straight line, or a zigzag pattern.

G.4.1. EXAMPLE 168

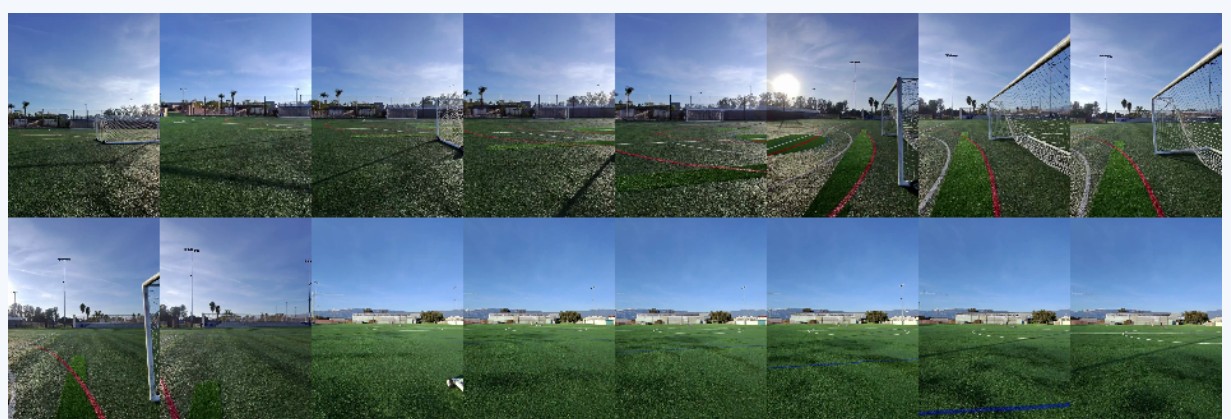

Reverse Route Plan 168

```
From my viewpoint at the end of the video, how can I go back to my starting point?

A. Turn around, go straight. Turn right, go straight, then turn left and continue go straight
B. Turn around, go straight. Turn left, go straight, then turn right and continue go straight
C. Turn around, go straight. Turn right, go straight, then turn right and continue go straight
D. Turn around, go straight. Turn left, go straight, then turn left and continue go straight
Answer: D. Turn around, go straight. Turn left, go straight, then turn left and continue go straight
```

**Gemini 3 Flash**
"options": A
"thinking_trace": To determine the return path, I first need to trace the forward path taken in the video.
1. Starting at 00:00.000, I am walking straight across the soccer field.
2. Around 00:11.000, I make a right turn.
3. I continue walking straight past a soccer goal on my right until about 00:21.500.
4. Around 00:22.000, I make a left turn.
5. I continue walking straight towards the center of the field until the end of the video.
To return to the starting point from the end of the video, I must reverse these steps:
1. Turn around (180 degrees) to face the direction I just came from.
2. Go straight (reversing the final segment).
3. Since my last turn was a left turn, I must now turn right to backtrack.
4. Go straight (reversing the middle segment).
5. Since my first turn was a right turn, I must now turn left to backtrack.
6. Go straight to reach the original starting point.
This sequence corresponds to Option A: Turn around, go straight. Turn right, go straight, then turn left and continue go straight.

**Gemini 3 Pro**
To determine the path back to the starting point, let's analyze the movement shown in the video:
1. **Start (00:00 - 00:11):** I begin facing a soccer goal on my right and a fence/building area ahead. I walk forward slightly and then turn left.
2. **First Turn (00:11 - 00:12):** I turn left, now facing the sun and the length of the field.
3. **Movement (00:12 - 00:22):** I walk straight along the sideline, passing the soccer goal on my right.
4. **Second Turn (00:22 - 00:23):** I turn right, facing across the width of the field towards a large building and mountains in the distance.
5. **Movement (00:23 - 00:35):** I walk straight across the field towards the center circle.
6. **End (00:35.500):** I am standing near the center circle, facing the building.
To reverse this path and return to the start:
1. **Turn around:** I need to face the opposite direction from where I am currently looking.
2. **Go straight:** This retraces the path from the center circle back to the sideline where I turned.
3. **Turn left:** Since I originally turned right to face the center, reversing this means turning left to face back down the sideline.

4. **Go straight:** Walk back along the sideline towards the initial corner.

5. **Turn right:** Since I originally turned left to start walking down the sideline, I need to turn right to face the original starting orientation (looking at the fence/goal area).

6. **Go straight:** Walk the short distance back to the exact starting spot.

Comparing this to the options:* Option A: Turn around, go straight. Turn right... (Incorrect turn direction).* Option B: Turn around, go straight. Turn left, go straight, then turn right and continue go straight. (Matches the reverse path: Turn around -¿ walk back across field -¿ Turn left at sideline -¿ walk down sideline -¿ Turn right to face original start).* Option C: Turn around, go straight. Turn right... (Incorrect turn direction).* Option D: Turn around, go straight. Turn left... then turn left... (Incorrect final turn).

Therefore, the correct sequence is to turn around, go straight back to the sideline, turn left to walk along it, and then turn right to face the original starting position.

"options": B

"thinking_trace": The video shows a path where I walk straight, turn left, walk straight for a while, turn right, and walk straight again to the end point. To return, I must reverse these actions in opposite order. First, I turn around (180 degrees). Then I walk straight, retracing the last leg of the journey. At the point where I made the second turn (which was a right turn), I must now turn left. Then I walk straight, retracing the middle leg. Finally, at the point where I made the first turn (which was a left turn), I must turn right and walk straight to the original starting spot. This corresponds to the sequence: Turn around, go straight, turn left, go straight, turn right, go straight. Option B matches this description.

**Gemini 2.5 Pro**

"options": A

"thinking_trace":

1. **Analyze my starting position and initial movement:** The video begins with me standing on an artificial turf soccer field. I am looking across the field towards a goal in the distance. There is another goal to my right. From 0:00 to 0:11, I walk forward.

2. **Analyze the first turn:** At 0:11, I turn my head and body to the right, panning across the field. The sun, which was to my right, is now in front of me. This confirms a right turn.

3. **Analyze the second movement:** From 0:12 to 0:19, I walk straight, parallel to the goal that is now on my right side.

4. **Analyze the second turn:** At 0:19, I pivot my body to the left and start walking away from the goal line, towards the center of the field. This is a left turn.

5. **Analyze the final movement:** From 0:20 to the end of the video, I walk straight towards the center of the field. The video ends with me facing the center circle and the far side of the field.

6. **Summarize the forward path:** * Go straight. * Turn right. * Go straight. * Turn left. * Go straight.

7. **Determine the reverse path:** To return to my starting point from my final position, I must reverse these actions in reverse order. * First, I need to turn around (180 degrees) to face the direction I came from. * Then, I walk straight back, reversing the final "Go straight" action. * Next, I need to reverse the Ťurn left: The opposite of a left turn is a **right turn**. * Then, I walk straight back, reversing the second G̈o straightäction. * Next, I need to reverse the "Turn right". The opposite of a right turn is a **left turn**. * Finally, I walk straight back to my original starting point, reversing the first "Go straight" action.

8. **Construct the full reverse instruction:** Turn around, go straight. Turn right, go straight, then turn left and continue go straight.

9. **Match with the given options:** * 'A': 'Turn around, go straight. Turn right, go straight, then turn left and continue go straight.' - This matches my derived reverse path. * 'B': 'Turn around, go straight. Turn left, go straight, then turn right and continue go straight.' - Incorrect. The first turn on the way back should be right. * 'C': 'Turn around, go straight. Turn right, go straight, then turn right and continue go straight.' - Incorrect. The second turn on the way back should be left. * 'D': 'Turn around, go straight. Turn left, go straight, then turn left and continue go straight.' - Incorrect. Both turns are wrong.

Therefore, option A correctly describes the path back to the starting point.

**GPT-5.2** "options": A

"thinking_trace": From the start I moved along the field, then made a right turn to go along the side near the goal, and later made a left turn to head toward the center/building. Reversing that route from the end: turn around and go straight, then turn right and go straight, then turn left and continue straight to reach the starting point.

**Qwen3-VL 235B** "options": A

"thinking_trace": To determine the path back to the starting point, analyze the video's movement: the starting point is near the soccer goal visible in early frames, and the end point is on a field with a circular line facing a building. The correct path requires reversing the trajectory. From the end, turning around (opposite the current view) and proceeding straight retraces the main path. Subsequent turns must counteract the forward-direction turns: turning right (instead of left) and then left (instead of right) aligns with retracing the route past goals and field markings. Option A matches this sequence: turn around, go straight, turn right, go straight, then turn left to reach the starting goal.

**Qwen3-VL 32B**

"options": B

"thinking_trace": To determine the path back to the starting point, we reverse the movement sequence. The video shows the camera moving from the start, turning in certain directions. The correct reverse path would involve turning around first, then following the opposite of the original turns. Option B describes a sequence where after turning around and moving straight, a left turn (opposite of a previous right turn in the forward path) is followed by a right turn to realign with the start, which matches the required reversal of directions observed in the video frames.

## Qwen3-VL 30B

"options": A

"thinking_trace": To determine the path back to the starting point, we analyze the movement from the starting frame (0.2s) to the end frame (35s). The starting point is near the right-side goal. In the video, the camera moves leftward along the field, then across the central area. To return, we reverse this path: 1) Turn around (face the direction of the starting point), 2) Go straight, 3) Turn right (to align with the right-side goal area), 4) Go straight, 5) Turn left (to reach the exact starting spot near the goal). This matches option A's sequence of "Turn around, go straight. Turn right, go straight, then turn left and continue go straight."

## Qwen2.5-VL 72B

"options": B

"thinking_trace": To return to the starting point from the viewpoint at the end of the video, I need to retrace my steps. Initially, I moved forward and then turned left, followed by another left turn. Therefore, to go back, I should first turn around and go straight to undo the last movement. Then, I need to turn left to align with the previous path and go straight. After that, I should turn right to correct the direction and continue going straight to reach the starting point. This sequence matches option B.

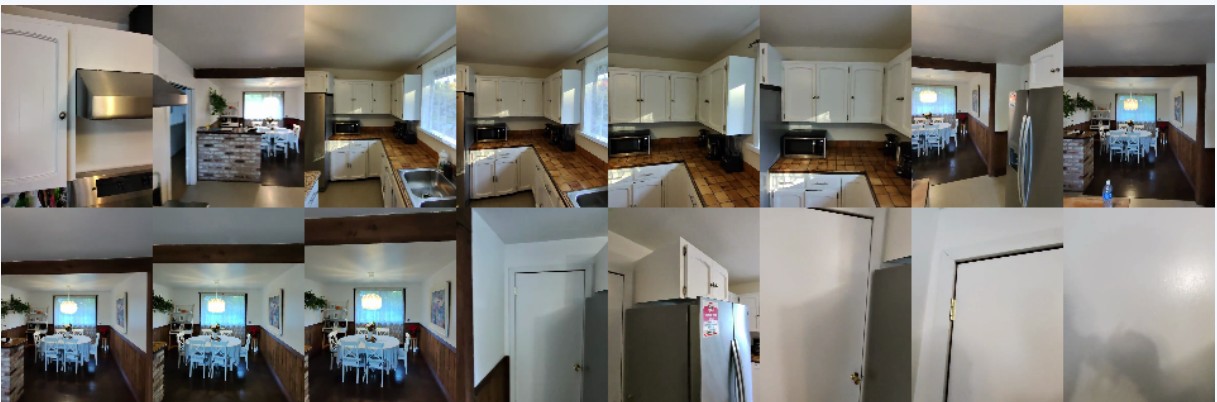

**Reverse Route Plan 196**

From my viewpoint at the end of the video, how can I go back to my starting point?

A. Turn around, go straight. Turn left, go straight, then turn left and continue go straight
B. Turn around, go straight. Turn right, go straight, then turn right and continue go straight
C. Turn around, go straight. Turn left, go straight, then turn right and continue go straight
D. Turn around, go straight. Turn right, go straight, then turn left and continue go straight

Answer: C. Turn around, go straight. Turn left, go straight, then turn right and continue go straight

**Gemini 3 Flash**

The provided video starts with a first-person view of a kitchen, specifically focusing on a white cabinet and a stainless steel range hood above a stove (00:00.000). As the camera pans to the right, it reveals an opening to a dining area (00:01.500) and then a stainless steel refrigerator (00:02.500). Further panning to the right shows more of the kitchen, including a microwave, a tiled countertop, and a sink (00:03.500 - 00:09.000).

The person then moves from the kitchen area into the dining room (00:10.500), which contains a white table and chairs. After looking around the dining room, the person turns around (00:18.500), exits the dining room, and turns toward a white door located next to the refrigerator (00:19.500 - 00:27.000), where the video ends.

To return to the starting point (the stove/cabinet area) from the final position at the door:

1. **Turn around**: The person must turn away from the door to face back into the kitchen area.
2. **Go straight**: Move past the refrigerator toward the center of the kitchen.
3. **Turn right**: To align with the path toward the dining room opening or to navigate the kitchen's layout as seen in the initial pan.
4. **Go straight**: Move toward the area between the dining room and the stove.
5. **Turn left**: This turn will orient the person to face the stove and the white cabinets where the video began.

This sequence of movements corresponds to the directions provided in option D.

options: {'A': 'Turn around, go straight. Turn left, go straight, then turn left and continue go straight.', 'B': 'Turn around, go straight. Turn right, go straight, then turn right and continue go straight.', 'C': 'Turn around, go straight. Turn left, go straight, then turn right and continue go straight.', 'D': 'Turn around, go straight. Turn right, go straight, then turn left and continue go straight.'}

"options": D

"thinking_trace": The video starts at the stove/cabinet area. It pans right to show the dining room opening, then the fridge, then the sink. The person walks into the dining room, turns around, exits, and turns toward a door near the fridge. To go back: from the door, turn around and go straight past the fridge. To reach the stove, which was to the left of the dining room opening (from the kitchen's perspective), the person would need to turn right toward the opening/center area and then turn left to face the stove. This matches the 'Turn right, then turn left' sequence in Option D.

**Gemini 2.5 Pro**

"options": D

"thinking_trace": 1. **Analyze the starting and ending points:** The video starts with me facing a stainless steel stove and range hood, with white cabinets to the left. The video ends with me standing directly in front of a plain white door.

2. **Map the spatial layout:** By observing the pan-around shot, I can construct a mental map of the room. * The starting point is the stove * To the right of the stove area is a kitchen island and an opening to a dining room. * Opposite the stove and island is a wall with a stainless steel refrigerator, and to the left of the refrigerator is the white door (the ending point). * The counter continues from

the refrigerator area and turns a corner to the right, where a sink is located under a window. The kitchen has an L-shape or U-shape layout with a central island.

3. **Determine the path taken from start to end:** The most logical path for me to have taken is from the stove to the door. This would involve: * Starting at the stove. * Turning right to face the main kitchen area. * Walking forward, likely passing the kitchen island. * Approaching the wall with the refrigerator and door. * Turning left to face the door. * Walking forward to the door. * The path is essentially: (From stove) Turn Right -¿ Go Straight -¿ Turn Left -¿ Go Straight -¿ Arrive at door.

4. **Reverse the path to find the way back:** To get from the ending point (the door) back to the starting point (the stove), I must reverse these actions in reverse order. * **Start at the end point:** I am facing the door. * **Step 1: Reverse the last action.** The last action was walking forward to the door. The reverse is to turn around (180 degrees) and walk straight away from the door. This corresponds to "Turn around, go straight." * **Step 2: Reverse the second-to-last action.** The action before that was turning left to face the door. The reverse is to turn right. * **Step 3: Reverse the action before that.** This was walking straight past the island. The reverse is to walk straight again. This corresponds to "Turn right, go straight." * **Step 4: Reverse the first action.** The first action was turning right from the stove. The reverse is to turn left, which will make me face the stove again. * **Step 5: Walk to the stove.** This corresponds to "then turn left and continue go straight."

5. **Combine the reversed steps:** The full sequence to return to the starting point is: Turn around, go straight, turn right, go straight, turn left, go straight.

6. **Match the reversed path to the given options:** * * 'A': Turn around, go straight. Turn left, go straight, then turn left and continue go straight. (Incorrect) * 'B': Turn around, go straight. Turn right, go straight, then turn right and continue go straight. (Incorrect) * 'C': Turn around, go straight. Turn left, go straight, then turn right and continue go straight. (Incorrect) * 'D': Turn around, go straight. Turn right, go straight, then turn left and continue go straight. (This matches the derived reverse path).

Therefore, option D correctly describes the path back to the starting point.

**GPT-5.2**
"options": A
"thinking_trace": At the end I'm facing a white door beside the fridge. To return to where I started (by the stove/range hood), I need to turn around into the kitchen, go straight to the open area, then make two left turns to head back along the counters to the range hood area.

**Qwen3-VL 235B**
"options": A
"thinking_trace": The video starts at the stove area and ends near a door. To return from the end point (door) to the start (stove), first turn around to face the dining area and go straight through it. Upon entering the kitchen, turn left to navigate toward the stove area. After going straight, another left turn aligns with the initial stove/cabinet position. This matches option A's sequence of turns.

**Qwen3-VL 32B**
"options": A
"thinking_trace": The video starts near the stove (kitchen area) and ends at a door. To return: 1. Turn around (face the direction from which you came, towards the dining area). 2. Go straight (through the dining area). 3. Turn left (to enter the kitchen, as the kitchen is to the left of the dining area). 4. Go straight (through the kitchen). 5. Turn left (to reach the starting point near the stove). This path matches option A.

**Qwen3-VL 30B**
"options": D
"thinking_trace": To determine the path back to the starting point, we analyze the video's sequence: 1. The starting point (frame 0-1) is near kitchen cabinets. 2. The end of the video (frame 26-27) shows a white door. 3. Tracing the movement: the journey from start to end involved moving through the kitchen, dining area, and toward the door. To reverse this, from the door: - Turn around (face away from the door, toward the dining area). - Go straight (toward the dining area). - Turn right (to navigate toward the kitchen). - Go straight (toward the kitchen). - Turn left (to align with the starting point near the initial kitchen cabinets). This matches option D.

**Qwen2.5-VL 72B**
"options": B
"thinking_trace": To return to the starting point from the viewpoint at the end of the video, I need to retrace my steps. The sequence of movements in the video shows a path that involves turning and moving straight multiple times. By analyzing the frames, it is clear that I need to turn around and go straight first, then make a right turn, continue straight, make another right turn, and finally continue straight to reach the starting point. This matches option B.

## G.5.1. EXAMPLE 53

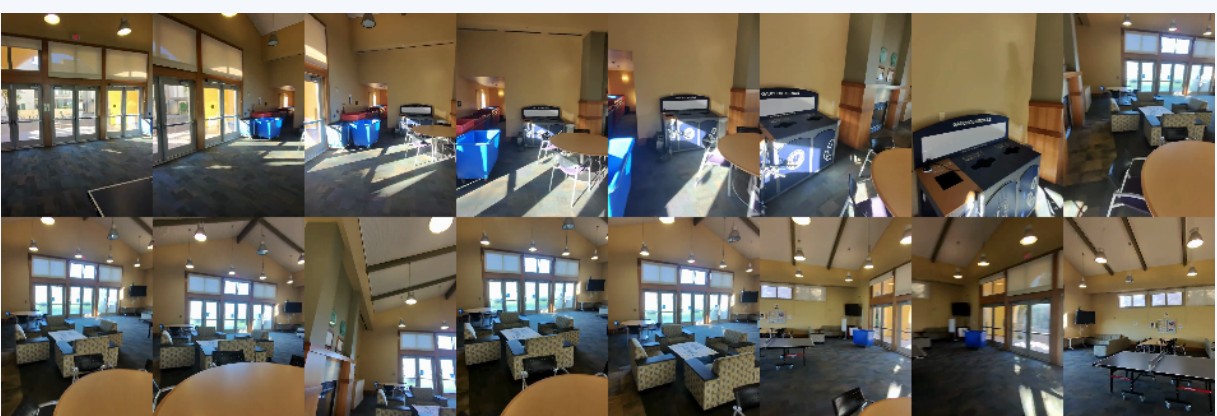

**Spatial Memory 53**

Which object changes between earlier and later in the video?

A. Scooter
B. Cart
C. Trash bin
D. Round table

Answer: B. Cart

---

**Gemini 3 Flash**

The object that changes between the earlier and later parts of the video is the scooter. In the earlier frames (e.g., 00:06.500 to 00:10.000), a silver kick scooter is clearly visible leaning against the "GAUCHOS RECYCLE" bin. However, when the camera returns to the same area later in the video (e.g., 00:21.500 to 00:22.500), the scooter is no longer present in that location. While other objects in the room also appear to have been moved or rearranged, the disappearance of the distinct scooter is a clear and notable change.

"options": A

"thinking_trace": The video shows a room from a first-person perspective. In the first half of the video (around 00:06.500 to 00:10.000), the camera pans past a "GAUCHOS RECYCLE" bin where a silver kick scooter is leaning. The camera then pans around the rest of the room, showing various lounge areas, tables, and a ping pong table. When the camera eventually pans back to the area with the recycle bin and carts (around 00:21.500 to 00:22.500), the scooter is no longer there. This indicates that the scooter is the object that changed (disappeared) between the earlier and later parts of the video.

---

**Gemini 3 Pro**

"options": A

"thinking_trace": The video starts by showing a room with large glass doors, blue recycling carts, and a round table. Around 00:02.000, the camera focuses on a recycling station labeled "GAUCHOS RECYCLE". Next to this station, there is a scooter parked. The scooter is clearly visible from 00:05.000 to 00:08.000. Later in the video, the camera pans around the room showing seating areas and tables. Towards the end of the video, specifically around 00:21.000 and 00:22.000, the camera returns to view the area near the recycling station and the glass doors. In these later frames, the scooter that was previously parked next to the recycling station is no longer there. The blue carts, trash bins, and round tables remain in their positions or are consistent with the room's layout, but the scooter has been removed. Therefore, the object that changes (disappears) is the scooter.

---

**Gemini 2.5 Pro**

"options": A

"thinking_trace": The user wants me to identify an object that changes its position or state during the video.

1. **Initial Observation (00:00 - 00:03):** I am in a large common room. I see glass doors, large wheeled carts (blue and red), a round table with chairs, and a recycling/trash bin station.

2. **Mid-Video Observation (00:04 - 00:08):** I move closer to the recycling bin. A silver and black scooter is clearly visible leaning against the front of the blue "GAUCHOS RECYCLE" bin.

3. **Panning Away (00:09 - 00:20):** I pan away from the recycling bin area, looking around the rest of the room, which includes a seating area with couches, a fireplace, and a ping pong table.

4. **Final Observation (00:21 - 00:24):** I pan back across the room. When the area with the recycling bin comes back into view, the scooter that was previously leaning against it is no longer there. The recycling bin itself, the carts, and the round tables are all in the same positions as before.

**Conclusion:** The scooter is present in the middle of the video but is absent at the end. This is a change. The carts, trash bin, and round table do not change their positions. Therefore, the scooter is the object that changes.

---

**GPT-5.2**
"options": A
"thinking_trace": A scooter becomes visible later in the video (near the recycling station), whereas it is not present in the earlier frames.

---

**Qwen3-VL 235B**
"options": C
"thinking_trace": Analyzing the frames, the trash bin (recycling station labeled 'GAUCHOS RECYCLE') is visible in multiple frames. Comparing earlier and later segments, the trash bin's presence and position change relative to the camera's movement, indicating it is the object that changes between earlier and later in the video. Other options like scooter, cart, and round table do not show a significant change in their state or position as observed in the frames.

---

**Qwen3-VL 32B**
"options": A
"thinking_trace": In the video frames, a scooter is visible near the 'GAUCHOS RECYCLE' bin in some earlier frames (e.g., frames 7-9) but is not present in later frames (e.g., frames 22-24). The other options (Cart, Trash bin, Round table) remain consistently visible or unchanged throughout the video sequence.

---

**Qwen3-VL 30B**
"options": D
"thinking_trace": Änalyzing the frames: A (Scooter) is not visible in any frame. B (Cart) shows blue carts in early frames (e.g., frame 3) and later frames (e.g., frame 24), so no change. C (Trash bin) remains visible (e.g., 'GAUCHOS RECYCLE' bin in frames 6-10 and later). D (Round table) is visible in multiple segments: seen near the recycling bin (frame 6), in the mid-section (frames 11-12), and later (frames 20-24). While the cart and trash bin remain consistent, the round table's position or presence in different contexts suggests it is the object that changes in terms of spatial context or visibility across the sequence.

---

**Qwen2.5-VL 72B**
"options": C
"thinking_trace": In the earlier part of the video, there is a blue trash bin visible near the entrance. As the video progresses and the camera moves around the room, the blue trash bin is no longer in view. This indicates that the position or visibility of the trash bin has changed between the earlier and later parts of the video.

### Spatial Memory 65

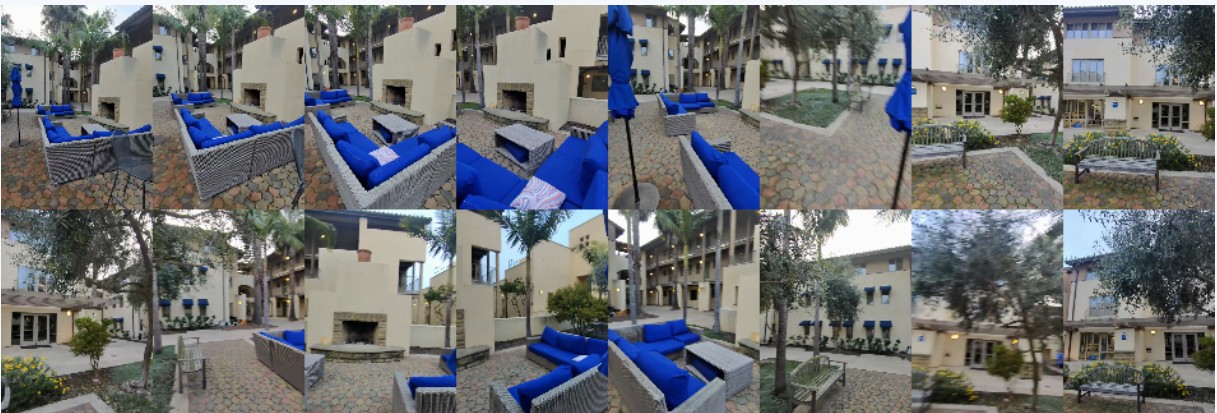

Which object changes between earlier and later in the video?

A. Outdoor coffee table
B. Seat cushion
C. Metal patio chair
D. Throw pillow

Answer: A. Outdoor coffee table

---

**Gemini 3 Flash**
"options": D
"thinking_trace": To determine which object changes between the earlier and later parts of the video, I will examine each option against the provided frames.
1. **Outdoor coffee table**: This is visible in frames like 00:01.500, 00:03.000, 00:11.000, and 00:14.000. It remains in the same position and appearance throughout the video.
2. **Seat cushion**: The blue cushions on the wicker sofas are visible throughout the video (e.g., 00:01.500, 00:11.000, 00:14.000). They do not appear to change.
3. **Metal patio chair**: A metal chair is visible in the bottom right of the frame at 00:00.000 and 00:00.500. As the camera moves, it goes out of view. There is no clear evidence of it c̈hangingöther than the camera moving away from it.
4. **Throw pillow**: In frame 00:02.500 and 00:03.000, a patterned throw pillow is clearly visible on the blue sofa. However, in later frames where the same sofa is visible, such as 00:11.000, 00:13.500, and 00:14.000, the patterned throw pillow is no longer there. This indicates a clear change in the scene.
Based on this analysis, the throw pillow is the object that changes (disappears) between the earlier and later parts of the video.

---

**Gemini 3 Pro**
The object that changes between earlier and later in the video is the throw pillow.
"options": D
"thinking_trace": In the beginning of the video (00:00.000 to 00:03.000), there is a patterned throw pillow on the blue sofa cushion closest to the camera. As the camera pans around the courtyard and returns to view the same sofa area later in the video (around 00:13.500 and onwards), the throw pillow is no longer present on the sofa. The other objects like the outdoor coffee table, seat cushions, and metal patio chair remain consistent in their presence or absence relative to the camera angle, but the pillow specifically disappears from its original location.

---

**Gemini 2.5 Pro**
"options": D
"thinking_trace": The user wants me to identify an object that changes its state or position during the video.
1. I first examine the scene at the beginning of the video (00:00-00:02). I see a patio with outdoor furniture. Specifically, I notice a large L-shaped wicker sofa with blue cushions. In the corner of this sofa, there is a single decorative throw pillow with a pink, purple, and white swirly pattern.

2. I then scan through the rest of the video, paying attention to the objects mentioned in the options. * The outdoor coffee table remains in the same position. * The blue seat cushions on the sofas remain in their places. * The metal patio chair, partially visible at the start, is out of frame for most of the video, but there's no indication it has changed.
3. I look back at the L-shaped sofa in the later parts of the video. At timestamp 00:11 and again at 00:13, I can see the corner of the sofa where the throw pillow was previously located. The pillow is no longer there.
4. Comparing the state of the sofa at the beginning (e.g., 00:02) with its state later on (e.g., 00:11), the most noticeable change is the disappearance of the throw pillow.
5. Therefore, the throw pillow is the object that changes between earlier and later in the video.

**GPT-5.2**
"options": D
"thinking_trace": A patterned throw pillow becomes visible on the blue outdoor sofa in later frames, whereas it is not present/visible in the earlier frames.

**Qwen3-VL 235B**
"options": D
"thinking_trace": The throw pillow is visible in some frames (e.g., frame 3) but not in others later in the video, indicating a change. Other options like the outdoor coffee table, seat cushion, and metal patio chair remain consistent in appearance and presence across the frames.

**Qwen3-VL 32B**
"options": D
"thinking_trace": In the video frames, a throw pillow is visible on the blue couch in earlier frames (e.g., frame 3 and 4) but is not present in later frames (e.g., frame 14 and 15). The other options (outdoor coffee table, seat cushion, metal patio chair) remain consistent in their presence or appearance throughout the video. Therefore, the throw pillow is the object that changes.

**Qwen3-VL 30B**
"options": D
"thinking_trace": Analyze each option:
n- A: Outdoor coffee table is consistently visible across frames.
n- B: Seat cushion (blue) remains unchanged on the couch.
n- C: Metal patio chair appears in some frames but does not undergo a visible change (it's present/repeatedly visible but not altered).
n- D: A throw pillow (with a striped pattern) appears on the couch in some frames (e.g., frame 3.2) but is absent in others (e.g., frame 1.2). Thus, its presence/absence changes between frames.
nOnly D shows a change in visibility between earlier and later frames."

**Qwen2.5-VL 72B**
"options": D
"thinking_trace": The throw pillow is visible on the seat cushion in the earlier part of the video. However, as the video progresses and the camera pans around, the throw pillow is no longer visible on the seat cushion. This indicates a change in the presence or position of the throw pillow between the earlier and later parts of the video.

**G.6.** Spatial Affordance

G.6.1. EXAMPLE 105

**Gemini 3 Flash**
"options": B
"thinking_trace": The video shows the person walking towards a Pepsi vending machine. By the end of the video (around 00:08.500 to 00:12.500), the person has stopped walking and extends their right arm towards the vending machine's control panel. The hand is clearly within reaching distance of the buttons while the camera (representing the person's head/body) remains stationary. This indicates that the buttons can be reached using only the arm without moving the feet further.

**Gemini 3 Pro**
"options": B
"thinking_trace": The video shows the person walking up to a Pepsi vending machine. As the person gets closer, the camera angle shifts to focus on the control panel of the machine. At the end of the video (around 00:08.500 onwards), the person extends their right arm towards the machine. The hand is clearly visible reaching out and is very close to the buttons and the coin slot area. The perspective suggests that the person is standing right in front of the machine, within arm's reach. The hand is extended and appears to be in a position to interact with the machine's interface. Therefore, it is possible to reach forward and touch the buttons using only the arm without moving the feet from that final position.

**Gemini 2.5 Pro**
"options": B
"thinking_trace": The video begins with me standing at a distance from a Pepsi vending machine. I then walk towards it. By the end of the video, I am standing directly in front of the machine. My hand enters the frame, reaching out towards the control panel where the buttons and payment slots are located. Given my close proximity to the machine and the position of my outstretched hand, it is clear that I can easily reach and touch the buttons without needing to take another step.

**GPT-5.2**
"options": B
"thinking_trace": In the later frames you are standing directly in front of the vending machine, and your outstretched hand reaches the area next to the button/keypad panel, indicating you can touch the buttons without stepping forward.

**Qwen3-VL 235B**

"options": B

"thinking_trace": In the video, the person's hand is shown reaching toward the vending machine buttons. The distance between the person and the machine appears close enough that extending the arm without moving the feet would allow contact with the buttons. Therefore, the answer is 'Yes'.

**Qwen3-VL 32B**

"options": B

"thinking_trace": In the 10th frame, the person's hand is extended towards the vending machine. The hand appears close enough to the machine's control panel to touch the buttons without needing to move their feet. This suggests that the person can reach the buttons with just their arm. Therefore, the correct answer is B.

**Qwen3-VL 30B** "options": B

"thinking_trace": In the video, the person moves towards the vending machine, and the hand extends towards the buttons. The frames show the hand reaching the vending machine's button area without needing to move feet, as the distance is covered by arm extension. Thus, the answer is B.

**Qwen2.5-VL 72B**

"options": B

"thinking_trace": The video shows that the person's hand is able to reach the vending machine buttons without any indication of moving their feet. The shadow and the position of the hand suggest that the person is standing close enough to touch the buttons with just an arm movement.

## Spatial Affordance 147

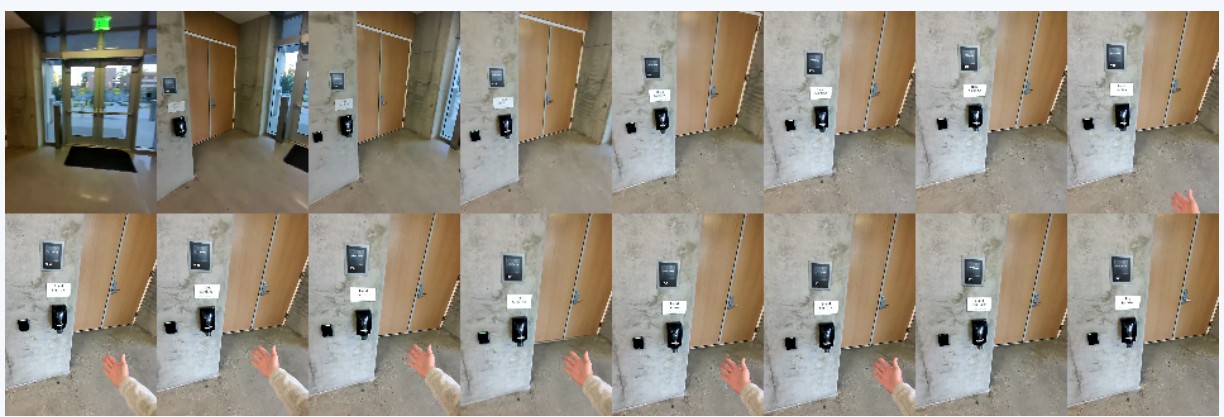

Based on the video, can I reach the hand sanitizer dispenser using only my arm after taking one small
    step leftward, without leaning or using anything?

A. Yes
B. No

Answer: B. No

---

**Gemini 3 Flash**

"options": A

"thinking_trace": In the video, the person is standing a short distance away from a wall where a hand sanitizer dispenser is mounted.
The dispenser is located to the left of the person's current central line of sight. By taking one small step to the left, the person would
be positioned directly in front of the dispenser. Given the proximity shown in the frames (especially around 00:03.000 to 00:06.000
where the hand is visible), the dispenser is well within a normal arm's reach from that position without needing to lean forward.

---

**Gemini 3 Pro**

"options": A

"thinking_trace": The video shows me standing in front of a wall with a hand sanitizer dispenser mounted on it. Initially, I am a few
steps away. As the video progresses, I move closer to the wall and the dispenser. By the end of the video (around frame 00:03.000
onwards), I extend my right arm towards the dispenser. My hand is clearly visible in the foreground, reaching out. Although I don't
touch it in the video, the perspective and the extension of my arm suggest I am within reaching distance. If I were to take one small
step to the left (towards the dispenser) from my final position, I would be even closer and directly in front of it, making it easily
reachable with just an arm extension without needing to lean excessively. Therefore, the answer is Yes.

---

**Gemini 2.5 Pro**

"options": A

"thinking_trace": 1. At the beginning of the video, I am standing in a hallway, facing a concrete wall with a hand sanitizer dispenser
on it.
2. At 00:03, my right hand enters the frame, extended towards the dispenser.
3. My hand is a noticeable distance away from the dispenser, suggesting I am standing a few feet away from the wall.
4. The question asks if I can reach the dispenser after taking one small step leftward.
5. Taking a step leftward would move my body closer to the wall where the dispenser is located.
6. Given the initial distance shown by my outstretched arm, a single small step would likely be sufficient to close the remaining gap,
allowing my arm to reach the dispenser without needing to lean or use any other aids.

---

**GPT-5.2**

"options": A

"thinking_trace": The dispenser is mounted on the concrete pillar very close to my position; my outstretched hand already appears
near it in the later frames. A small step left would bring my body closer/aligned with the pillar, making it reachable with just my arm

---

without needing to lean or use anything.

**Qwen3-VL 235B** "options": A
"thinking_trace": In frames 4-7, a hand is extended toward the hand sanitizer dispenser, indicating the dispenser is within arm's reach. Taking one small step leftward (as implied by the movement toward the dispenser's location) allows the arm to reach the dispenser without leaning or using additional tools, as shown by the hand's proximity to the dispenser in the video.

**Qwen3-VL 32B**
"options": A
"thinking_trace": In the video, the camera moves left from the initial position (facing the entrance doors) to reach the hand sanitizer dispenser. The frames show that after a small leftward movement, the hand sanitizer is within arm's reach. The visible hand in the later frames confirms that the arm can reach the dispenser without needing to lean or use additional support, as the distance is covered by the arm's natural reach after the small step. Thus, the answer is 'Yes'.

**Qwen3-VL 30B**
"options": A
"thinking_trace": In the frames, the hand sanitizer dispenser is on the wall. After a small leftward step, the hand reaches towards the dispenser, indicating the arm can reach it without leaning or additional support. Thus, the answer is Yes.

**Qwen2.5-VL 72B**
"options": A
"thinking_trace": The hand sanitizer dispenser is within arm's reach after taking one small step leftward. There is no need to lean or use anything else to reach it.

