# OpenReview forum: "SAW-Bench: Learning Situated Awareness in the Real World"
_ICML.cc/2026/Conference — ICML 2026 spotlight_

### Official Review · Reviewer_X2GW · 2026-03-12

**Soundness:** 3
**Presentation:** 3
**Significance:** 3
**Originality:** 3
**Overall Recommendation:** 5
**Confidence:** 4

**Summary:**

This paper introduces SAW-BENCH, a real-world situated reasoning benchmark that evaluates various aspect of situation awareness of current Large Foundation Models. The benchmark proposes a novel perspective of studying the relation between the agent and the environment rather than the relation inside the environment itself. Experiments with different models and analysis of the failure modes provide useful insight and inspire future research.

**Compliance With Llm Reviewing Policy:**

Affirmed.

**Final Justification:**

Overall I think most of the concerns are addressed and I'll keep my original positive score 5 (Accept).

**Key Questions For Authors:**

1. How is the Camera intrinsics set? Will the model's performance be affected if the provided video is recorded using a different set of intrinsics? An example is that in Spatial Affordance, changing the intrinsics would affect the belief of whether an object is reachable.

**Limitations:**

See Weaknesses.

**Strengths And Weaknesses:**

Strengths:

1. The proposed benchmark is novel in terms of considering egocentric visual inputs with situational awareness

2. The paper is well-written, well-structured, providing useful insights on the failure modes of the current foundation models in situated reasoning.

3. The paper conducts rigorous data collection and experiments.


Weaknesses:

1. The fairness of evaluation on Reverse Route Plan category is unclear, since there might be multiple ways of going back to the starting point of the trajectory. Clarification on this issue is appreciated.

2. The scale of the dataset is limited (786 videos and 2071 QA pairs), hindering the study of whether finetuning the current foundation model would mitigate the issue discussed in the paper.

3. The data is unbalanced, data in Relative Direction category is much more than data in Self-Localization, making it unclear whether the models are indeed slightly better at Self-Localization than at Relative Direction.

---

> ### Author Rebuttal · Authors · 2026-03-31
>
> ### 1. Weaknesses
>
> > **W1**: The fairness of evaluation on Reverse Route Plan category is unclear, since there might be multiple ways of going back to the starting point of the trajectory. Clarification on this issue is appreciated.
>
> In SAW-Bench, Reverse Route Plan is not an open-ended path-planning task with many equally valid ways to return. Each example is instead generated from a predefined trajectory template with known turns and motion segments, and the answer choices are restricted to a small set of candidate reverse plans derived from that trajectory. During annotation, QA pairs are grounded in the recording protocol and trajectory metadata, reducing ambiguity and ensuring that the correct answer is the intended reversal of the executed trajectory rather than any plausible alternative path in the environment. We will make the recording protocol and option construction clearer in the revised manuscript.
>
> > **W2**: The scale of the dataset is limited (786 videos and 2071 QA pairs), hindering the study of whether finetuning the current foundation model would mitigate the issue discussed in the paper.
>
> We would like to clarify that SAW-Bench is designed primarily for evaluation and diagnosis, rather than as a training set for fine-tuning foundation models. Our goal is to assess whether current multimodal foundation models exhibit situated awareness in egocentric video and to diagnose their failure modes under a controlled benchmark setting. This design choice is consistent with many recent benchmarks intended mainly for evaluating model capabilities, rather than for benchmark-specific training or fine-tuning [1,2,3,4].
>
> [1] TOMATO, ICLR 2025
>
> [2] Think-in-Space, CVPR 2025
>
> [3] CharXiv, NeurIPS 2024
>
> [4] MMMU, CVPR 2024
>
> > **W3**: The data is unbalanced, data in Relative Direction category is much more than data in Self-Localization, making it unclear whether the models are indeed slightly better at Self-Localization than at Relative Direction.
>
> We agree that the imbalance between Self-Localization and Relative Direction makes direct comparison less straightforward. To test whether the observed gap is a sample-size issue, we performed two additional analyses across four representative models:
> bootstrap confidence intervals (10,000 trials) for the accuracy difference (Self-Localization − Relative Direction) by resampling within each category with replacement, and
> matched-size down-sampling, where we repeatedly sample (5,000 trials) a Relative Direction subset of the same size as Self-Localization.
>
> Across all four models, Self-Localization consistently outperforms Relative Direction by 7.37–19.72 points; the bootstrap confidence interval is fully above zero for three of the four models, and the down-sampling analysis supports the same trend in 98.86%–100% of trials. This suggests that the difference is unlikely to be explained solely by category-size imbalance, and we will clarify this in the revised manuscript.
>
> | Model | Self-Loc | Rel-Dir | Diff | 95% Bootstrap CI of Diff | % Self-Loc > Downsampled Rel-Dir |
> |------------------------|----------|---------|------|---------------------------|--------------------------------|
> | Gemini-3-Flash   | 48.50    | 41.13   | 7.37 | [-0.38, 14.99]            | 98.86%                         |
> | Gemini-3-Pro      | 50.00    | 38.61   | 11.39 | [3.73, 19.03]             | 100.00%                        |
> | GPT-5.2              | 45.50    | 25.78   | 19.72 | [12.20, 27.30]            | 100.00%                        |
> | Qwen3-235B      | 43.50    | 33.41   | 10.09| [2.53, 17.74]             | 99.92%                         |
>
> Here, “% Self-Loc > Downsampled Rel-Dir” denotes the fraction of trials (5,000 trials) in which Self-Localization accuracy exceeds that of the matched-size Relative Direction subset.
>
>
> ### 2. Questions
>
> > **Q1**: How is the Camera intrinsics set? Will the model's performance be affected if the provided video is recorded using a different set of intrinsics? An example is that in Spatial Affordance, changing the intrinsics would affect the belief of whether an object is reachable.
>
> For Spatial Affordance, each task and its associated ground truth label are defined by a pre-specified recording protocol (`Appendix C.3`), rather than by post hoc visual judgment. For example, for an object such as a water bottle, the camera wearer records controlled cases, including:
>
> 1. being able to reach the object,
> 2. taking a step backward such that the object is no longer reachable without leaning, and
> 3. moving leftwards/rightwards until the object is no longer reachable.
>
> Thus, the label is grounded in the physical recording setup and action feasibility of the wearer, not in the subjective interpretation of pure visual information. This is consistent with our broader annotation procedure, where QA pairs are grounded in predefined recording protocols and known trajectory metadata.

---

> > ### Author Rebuttal · Reviewer_X2GW · 2026-04-02
> >
> > I would like to thank the author for their thorough responses. They mostly address my concerns. I still have one clarification / follow-up regarding the intrinsics of the camera. I appreciate the rigor in recording the dataset, what I was trying to ask is whether the video provided to a reasoning model with different apparent object scale (which can be an outcome from different camera intrinsics), would affect the model's performance. Overall I think most of the concerns are addressed and I'll keep my original positive score 5 (Accept).

---

> > > ### Author Response · Authors · 2026-04-06
> > >
> > > Thank you for the clarification. In SAW-Bench, all videos are recorded using the same Ray-Ban Meta glasses, so the camera intrinsics (*e.g.*, focal length and FOV) are fixed across the benchmark. As a result, variation in apparent object scale caused by different intrinsics is not a confounding factor in our current dataset. We agree that robustness to changes in camera intrinsics is an important cross-device generalization question and a valuable direction for future work. We will clarify this point in the paper.

---

### Official Review · Reviewer_iU1y · 2026-03-12

**Soundness:** 3
**Presentation:** 4
**Significance:** 3
**Originality:** 3
**Overall Recommendation:** 4
**Confidence:** 3

**Summary:**

The paper introduces SAW-BENCH, a video benchmark to evaluate situated awareness in LLMs, i.e., the ability to reason about the spatial relations involving the subject of a video. The benchmark consists of 786 real-world videos captured from an egocentric perspective using the RayBan Meta glasses. Videos are then annotated with ~2k question-answer pairs that focus on different reasoning capabilities, ranging from trajectory reasoning to spatial and memory skills. The paper presents an evaluation of several open and closed source LLMs, showing a significant gap compared to human judgement and a substantial failure of blind and socratic models (i.e., models provided with just a textual caption of the video).

**Compliance With Llm Reviewing Policy:**

Affirmed.

**Final Justification:**

The rebuttal positively addressed my questions about the comparison with models trained specifically for spatial reasoning and the use of a variable number of input frames. I maintain my original positive evaluation of this paper.

**Key Questions For Authors:**

Minor questions:
1. How many frames are given to the LLM? This detail appears to be missing in the paper. A baseline with one or very few input frames could be a valuable comparison, together with the blind and socratic models.

2. Could some of the described failure cases be addressed by providing the models with additional information, such as object detections or estimated camera motion?

3. How do models trained specifically to improve spatial reasoning skills compare on the proposed benchmark? (See above)

**Limitations:**

yes

**Strengths And Weaknesses:**

Situated awareness and spatial reasoning are fundamental skills to understand human behavior in the real world and current models are clearly lacking in this regard. The paper differentiates from prior work by focusing on the egocentric perspective of the camera recorder, which is relatively underexplored and makes spatial reasoning particularly challenging due to the continuous movements and limited field of view. The analyses of the failure modes of the existing LLMs is very well curated and detailed, though some of the observed limitations of current models are not entirely surprising, and prior work has already highlighted weaknesses in spatial reasoning.

The paper is well presented and motivated. I do not see critical issues in the paper other than its somewhat limited scope.
Indeed, the main limitation of the paper is that it primarily focuses on evaluation and analysis, providing limited insights into how the identified limitations in spatial reasoning of current LLMs might be addressed.
For example, providing the model with estimates of the camera trajectory could improve route related reasoning, while a simple tracking / re-identification pipeline could help the model better track already seen objects. It would also be interesting how different prompts or chain of thought strategies could encourage more spatial aware reasoning.

Only “generalist” LLMs are evaluated, either open or proprietary. A few models have been proposed to explicitly tackle spatial reasoning, such as [1], [2] and [3] (this last work could be consider a concurrent work). Despite being training on different spatial reasoning tasks, it would be interesting to evaluate them on the proposed benchmark to assess their failure modes compared to the generalist models.

[1] Wu, Junfei, et al. "Reinforcing Spatial Reasoning in Vision-Language Models with Interwoven Thinking and Visual Drawing." NeurIPS 2025

[2] Wu, Diankun, et al. "Spatial-MLLM: Boosting MLLM Capabilities in Visual-based Spatial Intelligence." NeurIPS 2025

[3] Yang, Shusheng, et al. "Cambrian-s: Towards spatial supersensing in video." arXiv preprint arXiv:2511.04670 (2025).

---

> ### Author Rebuttal · Authors · 2026-03-31
>
> ### 1. Weaknesses
>
> > **W1**: Only “generalist” LLMs are evaluated, either open or proprietary. A few models have been proposed to explicitly tackle spatial reasoning, such as [1], [2] and [3] (this last work could be consider a concurrent work). Despite being training on different spatial reasoning tasks, it would be interesting to evaluate them on the proposed benchmark to assess their failure modes compared to the generalist models.
> [1] VILASR, NeurIPS 2025
> [2] Spatial-MLLM, NeurIPS 2025
> [3] Cambrian-S, ICLR 2026
>
> > **Q3**: How do models trained specifically to improve spatial reasoning skills compare on the proposed benchmark? (See above)
>
> Thank you for your suggestions. Evaluation results of Spatial-MLLM is attached below, with comparison to Gemini-3-Flash and GPT-5.2:
>
> | Model | Overall | Self-Localization | Relative Direction | Route Shape | Reverse Route Plan | Spatial Memory | Spatial Affordance |
> |----------------------|---------|---------|----------|---------|---------|---------|----------|
> | Spatial-MLLM   |  30.16 | 36.84 | 19.30 | 36.58 | 24.89 | 34.00 | 49.38 |
> | InternVL3 14B | 32.69 | 49.00 | 17.27 | 45.05 | 24.02 | 54.00 | 49.38 |
> | Gemini-3-Flash | 53.89 | 48.50 | 41.13 | 64.84 | 61.57 | 66.00 | 70.99 |
> | GPT-5.2            | 41.04 | 45.50 | 25.78 | 50.55 | 44.98 | 63.00 | 62.96 |
>
> These results suggest that, although Spatial-MLLM is specifically trained to improve spatial reasoning, it does not show clear improvements on situated awareness in SAW-Bench, and its performance is broadly comparable to the InternVL3 series rather than exceeding strong generalist models. This further supports the value of SAW-Bench as a testbed for assessing whether improvements from spatially specialized training transfer to observer-centric egocentric-video reasoning.
>
>
> ### 2. Questions
>
> > **Q1**: How many frames are given to the LLM? This detail appears to be missing in the paper. A baseline with one or very few input frames could be a valuable comparison, together with the blind and socratic models.
>
> We do specify the video sampling protocol in the caption of ``Table 2: unless otherwise noted, all models process videos sampled at 2 fps. In our evaluation, the number of frames varies by model due to model-specific input constraints.’’ Specifically, `LLaVA-series`, `InternVL2-8B`, and `InternVL3-8B/14B` take 32 frames as input, while `InternVL2-40B` and `InternVL3-38B` use 8 frames due to compute limitations. We also include a frame-sensitivity analysis in Table 8, which compares performance under different numbers of input frames. We will revise the main text to make this setup more explicit and easier to locate.
>
> We agree that a one-frame or very few-frame baseline could be a valuable comparison. We include additional results with one randomly sampled frame and two uniformly sampled frames, reported in the table below:
>
> | Model | Overall | Self-Localization | Relative Direction | Route Shape | Reverse Route Plan | Spatial Memory | Spatial Affordance |
> |---------|-------|----------------|----------------|-------------|-----------------|----------------|--------------------|
> | Blind LLM (GPT-5.2) | 31.34 | 38.00 | 23.02 | 36.63 | 24.02 | 38.00 | 54.32 |
> | Socratic (GPT-5.2)    | 31.34 | 40.50 | 20.62 | 41.58 | 24.02 | 32.00 | 50.62 |
> | 1 Frame (GPT-5.2)   | 28.92 | 44.50  | 14.87 | 38.64 | 21.40 | 38.00 | 54.32 |
> | 2 Frames (GPT-5.2) | 36.46 | 45.00 | 26.50 | 46.89 | 25.76 | 36.00 | 57.41 |
> | 2 FPS (GPT-5.2)      | 41.04 | 45.50 | 25.78 | 50.55 | 44.98 | 63.00 | 62.96 |
>
>
> > **Q2**: Could some of the described failure cases be addressed by providing the models with additional information, such as object detections or estimated camera motion?
>
> We believe the answer is partly yes. However, we would like to clarify that SAW-Bench is designed to evaluate **end-to-end situated awareness** from raw egocentric video alone, without access to external geometry, tracking, or state tracking information. This is intentional: our goal is to test whether current MFMs can infer observer-centric spatial structure directly from first-person visual input, rather than from auxiliary information such as object detections or estimated camera motion.
>
> That said, additional signals, such as estimated camera motion, object tracks, or depth estimates, could plausibly mitigate certain failure modes. For example, explicit camera-motion estimation may help with Route Shape, Relative Direction, and Reverse Route Plan, where models often conflate head rotation with translational movement. Our analysis suggests, however, that the main bottleneck is not simply a lack of object recognition, but **failure to maintain a coherent observer-centric representation over time**: models often confuse camera rotation with body movement and fail to preserve persistent world-state memory across frames.

---

> > ### Author Rebuttal · Reviewer_iU1y · 2026-04-04
> >
> > My concerns have been addressed and i will keep my positive score.

---

### Official Review · Reviewer_1oAW · 2026-03-13

**Soundness:** 2
**Presentation:** 3
**Significance:** 2
**Originality:** 2
**Overall Recommendation:** 4
**Confidence:** 4

**Summary:**

The authors present a benchmark consisting of 786 egocentric videos recorded with Ray-Ban Meta smart glasses, paired with 2,071 human-annotated multiple-choice questions. SAW-BENCH evaluates six distinct tasks: Self-Localization, Relative Direction, Route Shape, Reverse Route Plan, Spatial Memory, and Spatial Affordance. Their evaluation of 24 models shows that even the top-performing model (Gemini 3 Flash) lags behind human performance by roughly 32%. Key findings suggest models often confuse head rotation with physical movement and struggle with complex trajectories.

**Compliance With Llm Reviewing Policy:**

Affirmed.

**Final Justification:**

Thanks for the further clarification. All of my concerns have been addressed. Thus, I increase my score to 4.

**Key Questions For Authors:**

1. How does SAW-BENCH differentiate itself from the MSNN (Next-step Navigation) task in MSQA, which also requires reasoning about movement from a situated perspective?
2. Recent benchmarks like Ego3D-Bench[1] utilize multi-view outdoor data to evaluate absolute/relative distance and motion reasoning. Since SAW-BENCH relies on single-view egocentric video from Ray-Ban Meta glasses, how do the authors account for the lack of depth perception or multi-view consistency when evaluating 'Spatial Affordance'?
3. The paper identifies the 'non-visibility = non-existence' error. Did the authors experiment with providing models a 'Bird's Eye View' (BEV) map or a persistent spatial memory buffer? Without these, isn't it expected that frame-based MFMs will fail the 'Spatial Memory' tasks by design?

[1] Spatial Reasoning with Vision-Language Models in Ego-Centric Multi-View Scenes, ICLR 2026

**Limitations:**

1. The average video length in SAW-BENCH is roughly 27 seconds. This does not test "ultra-long" memory retention or temporal localization, challenges that are central to 2026 benchmarks like X-LeBench[1] and EgoGraph[2], which use hour-long or multi-day recordings.
2. Short clips allow models to rely on "keyframe shortcuts" (e.g., comparing the first and last frames) rather than maintaining a continuous, structured world-state over time.
3. The 15 scenes used are largely static. A limitation should be noted regarding the absence of dynamic obstacles (e.g., moving pedestrians, vehicles), which are essential for true situated "awareness" in realistic, high-entropy environments.

[1] X-LeBench: A Benchmark for Extremely Long Egocentric Video
[2] EgoGraph: Temporal Knowledge Graph for Egocentric Video Understanding

**Strengths And Weaknesses:**

### Strengths
1. Unlike many benchmarks that use synthetic or third-person data, this uses authentic egocentric video from wearable tech, providing a realistic "first-person" perspective.
2. The six tasks are well-defined and derived from cognitive science principles, covering diverse aspects of spatial intelligence from memory to physical action feasibility.
3. The benchmark uses a solid pipeline including manual review of videos for stability and multiple-human validation of ground-truth answers.

### Weaknesses
1. While the egocentric real-world focus is strong, the conceptual novelty is somewhat limited. The paper itself acknowledges SQA3D[1] as highly relevant, and the core tasks (localization, route planning) are established challenges in robotics and navigation.
2. The annotators are a small group of undergraduate and master's students. While they achieved high agreement, a broader demographic might provide a more robust "human baseline."
3. The paper's claim of addressing a "largely overlooked" gap in observer-centric relationships is weakened by the existence of MSQA[2]. While SAW-BENCH focuses on the egocentric video modality, MSQA already established the framework for reasoning about object existence, spatial relationships, and navigation (via the MSNN - Multi-modal Next-step Navigation task) grounded in a 3D environment.

[1] SQA3D: Situated Question Answering in 3D Scenes, ICLR 2023

[2] Multi-modal Situated Reasoning in 3D Scenes, NeurIPS 2024

---

> ### Author Rebuttal · Authors · 2026-03-31
>
> ### 1. Weaknesses
>
> > **W1**: While the egocentric real-world focus is strong, the conceptual novelty is somewhat limited. The paper itself acknowledges SQA3D as highly relevant, and the core tasks (localization, route planning) are established challenges in robotics and navigation.
>
> We agree that capabilities such as localization and route reasoning are well-established in robotics and navigation. Our novelty is not that these tasks are newly introduced, but that SAW-Bench benchmarks these capabilities in a real-world egocentric-video setting for MFMs. In particular, SQA3D operates on reconstructed 3D scene context, whereas SAW-Bench evaluates situated awareness directly from egocentric video, without explicit 3D reconstruction or global scene representations. This requires models to infer and maintain observer-centric spatial state from first-person visual experience, a setting that is especially relevant as MFMs are increasingly deployed on first-person devices such as AR/VR and smart glasses.
>
> > **W2**: The annotators are a small group of undergraduate and master's students. While they achieved high agreement, a broader demographic might provide a more robust "human baseline."
>
> We agree that a broader demographic could strengthen the human baseline. However, SAW-Bench mainly tests basic spatial understanding in egocentric video rather than subjective or open-ended reasoning. Since answers are grounded in the recording protocol (`Table 6`), we expect the human baseline to be relatively stable across annotator backgrounds.
>
> > **W3**: The paper's claim of addressing a "largely overlooked" gap in observer-centric relationships is weakened by the existence of MSQA. While SAW-BENCH focuses on the egocentric video modality, MSQA already established the framework for reasoning about object existence, spatial relationships, and navigation (via the MSNN - Multi-modal Next-step Navigation task) grounded in a 3D environment.
>
> We would like to clarify that our claim is that observer-centric situated awareness remains underrepresented in current MFM evaluation for realistic egocentric-video settings. While MSNN focuses on next-step navigation in a structured 3D environment, SAW-Bench evaluates a broader set of observer-centric capabilities directly from raw egocentric video, including spatial memory and spatial affordance. These are especially relevant for embodied MFMs on first-person devices such as smart glasses and AR/VR. We will clarify this distinction in the revision.
>
>
> ### 2. Questions
>
> > **Q1**: How does SAW-BENCH differentiate itself from the MSNN (Next-step Navigation) task in MSQA, which also requires reasoning about movement from a situated perspective?
>
> MSNN and SAW-Bench differ in both setting and scope. MSNN evaluates next-step navigation in a structured 3D environment, while SAW-Bench evaluates whether MFMs can maintain observer-centric spatial state from raw egocentric video, without explicit 3D reconstruction or global scene representations. Beyond route reasoning, SAW-Bench also includes Spatial Memory and Spatial Affordance, which probe object persistence and action feasibility from a first-person viewpoint. Thus, SAW-Bench targets a broader notion of situated awareness in realistic embodied settings, beyond next-step navigation.
>
> > **Q2**: Recent benchmarks like Ego3D-Bench utilize multi-view outdoor data to evaluate absolute/relative distance and motion reasoning. Since SAW-BENCH relies on single-view egocentric video from Ray-Ban Meta glasses, how do the authors account for the lack of depth perception or multi-view consistency when evaluating 'Spatial Affordance'?
>
> SAW-Bench evaluates whether MFMs can judge first-person action feasibility from raw egocentric video alone, without explicit depth, multi-view consistency, or 3D reconstruction. This matches realistic embodied deployment, where the input is often a single egocentric video. By contrast, Ego3D-Bench targets multi-view outdoor spatial reasoning with stronger geometric cues. In this way, the two benchmarks target different capabilities.
>
> > **Q3**: The paper identifies the 'non-visibility = non-existence' error. Did the authors experiment with providing models a 'Bird's Eye View' (BEV) map or a persistent spatial memory buffer? Without these, isn't it expected that frame-based MFMs will fail the 'Spatial Memory' tasks by design?
>
> We agree that a BEV map or persistent memory buffer would likely improve Spatial Memory. However, SAW-Bench is designed to test whether MFMs can maintain observer-centric spatial state from raw egocentric video alone, without bird’s-eye or global scene representations. Our goal is not to show how current models can succeed with privileged spatial structure, but to diagnose the gap between frame-level perception and persistent world-state reasoning in the standard real-world end-to-end setting.

---

> > ### Author Rebuttal · Reviewer_1oAW · 2026-04-03
> >
> > Thanks for the detailed response. Some of my concerns have been addressed. However, my main concern is still unsolved.
> >
> > The authors may have bias on SQA3D: "In particular, SQA3D operates on reconstructed 3D scene context, whereas SAW-Bench evaluates situated awareness directly from egocentric video, without explicit 3D reconstruction or global scene representations." In fact, SQA3D has a video input mode in evaluation (see figure 1's caption). A scene can be represented by an egoview video. SQA3D's data is also collected from the real-world videos. Thus, this weakens the claim of your paper's title.

---

> > > ### Author Response · Authors · 2026-04-06
> > >
> > > SQA3D is a valuable benchmark for situated question-answering in 3D scenes, and we acknowledge that it includes the corresponding egocentric view of the 3D scan. However, we would like to clarify the confusion about the key difference in benchmark construction: SQA3D is built on ScanNet scenes, where the videos were originally captured primarily for 3D reconstruction, *i.e.*, to support building the scene scan and 3D reconstruction, rather than to be explicitly designed to reflect **natural human movement trajectories within the environment**. As a result, these videos mainly provide scene coverage for reconstruction and annotation, instead of being purpose-recorded first-person trajectories specifically designed to evaluate dynamic situated awareness. Video comparisons are shown here (anonymous GitHub link):
> > > - SAW-Bench:
> > > [SAW-Bench 1](https://anonymous.4open.science/r/SAW-Bench-Videos-F08E/gifs/saw-bench/100_26.gif),
> > > [SAW-Bench 2](https://anonymous.4open.science/r/SAW-Bench-Videos-F08E/gifs/saw-bench/102_59.gif),
> > > [SAW-Bench 3](https://anonymous.4open.science/r/SAW-Bench-Videos-F08E/gifs/saw-bench/12_22.gif),
> > > [SAW-Bench 4](https://anonymous.4open.science/r/SAW-Bench-Videos-F08E/gifs/saw-bench/1_44.gif),
> > > [SAW-Bench 5](https://anonymous.4open.science/r/SAW-Bench-Videos-F08E/gifs/saw-bench/2_32.gif),
> > > [SAW-Bench 6](https://anonymous.4open.science/r/SAW-Bench-Videos-F08E/gifs/saw-bench/6_23.gif)
> > > - SQA3D:
> > > [SQA3D 1](https://anonymous.4open.science/r/SAW-Bench-Videos-F08E/gifs/sqa3d/scene0200_00.gif),
> > > [SQA3D 2](https://anonymous.4open.science/r/SAW-Bench-Videos-F08E/gifs/sqa3d/scene0201_00.gif),
> > > [SQA3D 3](https://anonymous.4open.science/r/SAW-Bench-Videos-F08E/gifs/sqa3d/scene0202_00.gif),
> > > [SQA3D 4](https://anonymous.4open.science/r/SAW-Bench-Videos-F08E/gifs/sqa3d/scene0213_00.gif),
> > > [SQA3D 5](https://anonymous.4open.science/r/SAW-Bench-Videos-F08E/gifs/sqa3d/scene0217_00.gif),
> > > [SQA3D 6](https://anonymous.4open.science/r/SAW-Bench-Videos-F08E/gifs/sqa3d/scene0246_00.gif),
> > >
> > > In addition, we highlight three key contributions of SAW-Bench in comparison with SQA3D:
> > >
> > > 1. **Dynamic rather than static definition of “situation”**: A key conceptual difference lies in how the “situation” is defined.
> > > . In SQA3D, the situation is anchored to a fixed position and orientation within a scene (Figure 3 I: Situation Identification). In contrast, in SAW-Bench, the situation is constructed dynamically over time through the observer’s movement. Because practical embodied perception and AR/VR require a persistent, continuously updated latent state, our benchmark evaluates whether models can maintain and update this evolving state during active exploration, rather than treating video as a description of the scene.
> > >
> > > 2. **QA designed inspired from cognitive science**: Our task design is directly inspired by the cognitive science framework of situational awareness, which emphasizes four core components: (1) navigation, (2) spatial updating, (3) spatial working memory, and (4) affordance. Evaluating these abilities requires temporally coherent first-person motion trajectories. By contrast, SQA3D (ScanNet) videos are primarily collected for scene reconstruction and coverage, rather than as natural egocentric trajectories designed to probe spatial updating, working memory, or affordance. As a result, question answering from a fixed, static viewpoint may capture spatial relations and scene-grounded reasoning around a specified viewpoint, but it does not fully test the dynamic mechanisms central to situated awareness. SAW-Bench is explicitly designed to evaluate these abilities.
> > >
> > > 3. **Purpose-built, self-recorded, in-the-wild data collection**: SQA3D is built by re-annotating existing indoor environments from ScanNet. In contrast, SAW-Bench is collected through a purpose-built pipeline designed from the ground up for evaluating situated awareness. We self-recorded all videos using Ray-Ban Meta Glasses, specifically to capture the aforementioned cognitive science aspects of situated awareness. The resulting dataset consists of real-world ego-centric videos spanning unconstrained **indoor** and **outdoor** environments, including settings such as courtyards, lawns, and plazas, which we believe better reflect practical embodied perception and AR/VR scenarios.

---

### Decision · Program_Chairs · 2026-04-30

**Decision:**

Accept (spotlight)

**Comment:**

The paper introduces SAW-Bench, a real-world egocentric video benchmark to evaluate situated spatial awareness in multimodal foundation models. Reviewers appreciated the use of RayBan Meta smart-glasses to collect real egocentric video footage, rather than relying on synthetic data, and recognized the relevance of the proposed benchmark and analysis. Given the positive consensus, I recommend acceptance.